# Nanog mediated control of TBX3-GATA6 circuitry in primitive endoderm differentiation of mESCs

Hao Wu[1,14], Ying Ye [2,14✉], Hongxia Dai [1], Peixin Chen[1], Tenghui Yang[1], Zhifang Li [3], Li Li [4], Chirag Parsania[5], Junjun Ding[6], Man Zhang[7], Erwei Zuo [3], Ulf Schmitz[8,9], Xi Chen [10✉], Zhexin Zhu [11✉] & Wensheng Zhang [1,12,13✉]

## Abstract

Cell fate decisions in the early embryo rely on reciprocal transcriptional networks that balance pluripotency with lineage commitment. NANOG is essential for directing the epiblast–primitive endoderm (PrE) fate choice, but the molecular mechanisms underlying its repressive activity remain incompletely understood. Here we show that NANOG partners with TBX3 and the PRC2 complex to maintain embryonic stem cell (ESC) identity by silencing PrE genes through newly identified distal enhancers. Loss of Nanog reduces PRC2-mediated repression of Gata6, initiating its expression independently of TBX3. Subsequent TBX3 upregulation enables its association with GATA6, driving a feed-forward programme that activates Gata6, Gata4 and Sox17 and promotes PrE differentiation. Thus, NANOG suppresses PrE fate not only by direct repression but also by preventing TBX3 from switching partners. These findings define a Nanog–Tbx3–Gata6 regulatory axis that integrates enhancer control, chromatin regulation and transcription factor redeployment to couple ESC maintenance with lineage commitment.

Keywords Nanog; Primitive Endoderm; Embryonic Stem Cells; PRC2 Complex
Subject Categories Development; Stem Cells & Regenerative Medicine

## Introduction

Embryonic stem cells (ESCs), derived from the inner cell mass (ICM) of early blastocysts, are defined by their capacity for indefinite self-renewal and differentiation into all three germ layers (Evans and Kaufman, 1981; Martin, 1981; Thomson et al, 1998). Pluripotency is governed by a network of transcription factors, signaling pathways, and chromatin regulators, including OCT4, SOX2, NANOG, and Polycomb complexes (Chambers and Tomlinson, 2009; Li and Belmonte, 2017; Tee and Reinberg, 2014; Ye et al, 2020). Among these, NANOG, a homeobox-containing transcription factor, is critical for ESC self-renewal and early embryonic development (Chambers et al, 2003; Mitsui et al, 2003). Overexpression of Nanog sustains mouse ESCs independently of LIF/STAT3, whereas its loss predisposes cells to differentiate toward the primitive endoderm (PrE) lineage.

Pre-implantation mouse development involves sequential cell fate decisions. The first segregates the ICM and trophectoderm, and the second partitions the ICM into epiblast (Epi) and PrE (Chowdhary and Hadjantonakis, 2022; Johnson and Ziomek, 1981; Saiz and Plusa, 2013). This second decision is orchestrated by mutually antagonistic factors, with NANOG marking Epi and GATA6 marking PrE (Chowdhary and Hadjantonakis, 2022; Gardner, 1982). Initially, ICM cells co-express NANOG and GATA6 at variable levels, reflecting a poised, uncommitted state (Dietrich and Hiiragi, 2007; Thompson et al, 2022). By E3.5, NANOG and GATA6 segregate into a "salt-and-pepper" distribution, giving rise to distinct Epi and PrE populations (Bessonnard et al, 2014; Chazaud et al, 2006). Within the PrE lineage, transcription factor activation follows a sequential hierarchy: GATA6 → SOX17 → GATA4 → SOX7 (Artus et al, 2011). Despite evidence of a mutually inhibitory relationship between NANOG and GATA6, the mechanisms by which NANOG regulates PrE differentiation remain incompletely defined(Frankenberg et al, 2011; Wamaitha et al, 2015).

Tbx3 has been implicated in both ESC self-renewal and differentiation, yet the mechanisms underlying its dual functions remain unclear (Ivanova et al, 2006; Zhang et al, 2019). Here, we show that in wild-type (WT) ESCs, TBX3 forms a complex with

[1]Cam-Su Genomic Resource Center, Medical College of Soochow University, Suzhou, China. [2]Department of Clinical Pathobiology and Immunological Testing, School of Medical Laboratory, Qilu Medical University, 255300 Zibo, China. [3]Shenzhen Branch, Guangdong Laboratory for Lingnan Modern Agriculture, Key Laboratory of Synthetic Biology, Ministry of Agriculture and Rural Affairs, Agricultural Genomics Institute at Shenzhen, Chinese Academy of Agricultural Sciences, Shenzhen, China. [4]College of Life Science and Technology, College of Biomedicine and Health, Huazhong Agricultural University, 430070 Wuhan, China. [5]Gene and Stem Cell Therapy Program Centenary Institute, The University of Sydney, Sydney, NSW, Australia. [6]RNA Biomedical Institute, Sun Yat-Sen Memorial Hospital, Zhongshan School of Medicine, Sun Yat-Sen University, Guangzhou, Guangdong, China. [7]Guangzhou National Laboratory, No. 9 XingDaoHuanBei Road, Guangzhou International Bio Island, 510005 Guangzhou, Guangdong Province, China. [8]Department of Biomedical Sciences and Molecular and Cell Biology, College of Public Health, Medical, and Veterinary Sciences, James Cook University, Townsville, QLD, Australia. [9]Centre for Tropical Bioinformatics and Molecular Biology, James Cook University, Cairns, QLD, Australia. [10]Shenzhen Key Laboratory of Gene Regulation and Systems Biology, School of Life Sciences, Southern University of Science and Technology, Shenzhen, China. [11]Institute of Health and Medicine, Hefei comprehensive national science center, 4090 Guanhai Road, Heifei, China. [12]Key Laboratory of Pesticide & Chemical Biology of Ministry of Education, Hubei Key Laboratory of Genetic Regulation and Integrative Biology, School of Life Sciences, Central China Normal University, 430079 Wuhan, China. [13]School of Life Sciences and Medicine, Shandong University of Technology, 255049 Zibo, China. [14]These authors contributed equally: Hao Wu, Ying Ye. ✉E-mail: yeying@qlmu.edu.cn; chenx9@sustech.edu.cn; zxzhu@ihm.ac.cn; zhangwensheng@suda.edu.cn

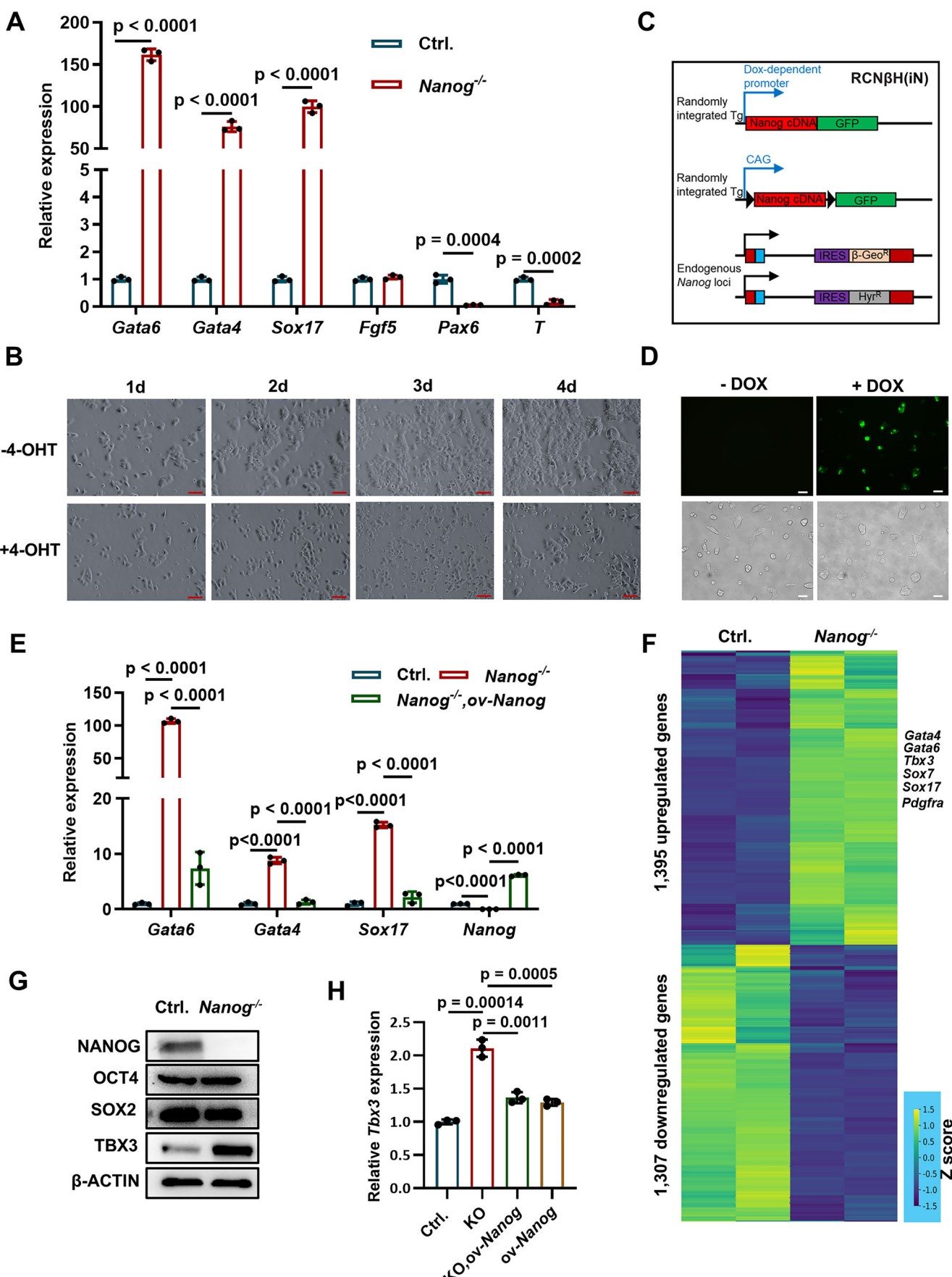

◄  **Figure 1.   *Nanog* represses PrE marker gene expression in ESCs.**

(A) qPCR analysis reveals the transcript levels of specified lineage-specific genes in RCNβH cells treated with (*Nanog*$^{-/-}$) and without 4-OHT (Ctrl.) for 4 days. Data are presented as mean ± SD ($n = 3$, biological replicates) *Gata6*, Row 1, $P = 2.38 \times 10^{-6}$; *Gata4*, Row 1, $P = 2.58 \times 10^{-5}$; *Sox17*, Row 1, $P = 1.65 \times 10^{-5}$. (B) Phase-contrast images depicting RCNβH ESCs without or with 4-OHT for 1 d, 2 d, 3 d, and 4 d in FBS/Lif culture condition consistent with those described by Chambers et al (Chambers et al, 2007). Scale bar, 50 μm. (C) Schematic diagram of RCNβH(iN) ESCs, where Doxycycline-inducible *Nanog* cDNA fused with GFP was integrated randomly into the genome of RCNβH ESCs. (D) Illustrative images showing RCNβH(iN) ESCs treated with and without 1 μg/ml of Doxycycline for 4 days. The addition of Doxycycline induced the expression of GFP-fused NANOG protein. Scale bar, 50 μm. (E) qPCR analysis depicts the expression levels of *Gata6*, *Gata4*, *Sox17*, and *Nanog* genes in RCNβH(iN) ESCs (Ctrl.), RCNβH(iN) ESCs treated with 4-OHT (*Nanog*$^{-/-}$), and RCNβH(iN) treated with 4-OHT along with Doxycycline (*Nanog*$^{-/-}$, ov-*Nanog*) over a 4-day period. Data are presented as mean ± SD ($n = 3$, biological replicates) *Gata6*, $P = 9.31 \times 10^{-7}$ (Row 1), $P = 3.24 \times 10^{-6}$ (Row 2); *Gata4*, $P = 5.85 \times 10^{-5}$ (Row 1), $P = 3.25 \times 10^{-5}$ (Row 2); *Sox17*, $P = 3.74 \times 10^{-5}$ (Row 1), $P = 2.69 \times 10^{-6}$ (Row 2); *Nanog*, $P = 6.05 \times 10^{-7}$ (Row 1), $P = 9.33 \times 10^{-8}$ (Row 2). (F) Heatmap illustrating the distinctly expressed genes identified through RNA-seq analysis in RCNβH (Ctrl.) and RCNβH ESCs treated with 4-OHT (*Nanog*$^{-/-}$) for 4 days. (G) Western blot analysis depicting the protein levels of NANOG, OCT4, SOX2, and TBX3 in RCNβH (Ctrl.) and RCNβH ESCs treated with 4-OHT (*Nanog*$^{-/-}$). β-ACTIN was used as a loading control. (H) qPCR assessment of *Tbx3* transcript levels in RCNβH(iN) ESCs (Ctrl.), RCNβH(iN) ESCs treated with 4-OHT (KO) or Doxycycline (ov-*Nanog*), and RCNβH(iN) treated with 4-OHT along with Doxycycline (KO,ov-*Nanog*) over a 4-day period. Data are presented as mean ± SD ($n = 3$, biological replicates). The unpaired two-tailed Student's *t* test was used for the statistical analysis in (A, E, H). Source data are available online for this figure. Source data are available online for this figure.

## Results

### Nanog regulates the PrE differentiation of ESCs

To elucidate how *Nanog* regulates PrE differentiation in ESCs, we utilized RCNβH ESCs (Appendix Fig. S1A; Chambers et al, 2007). Consistent with previous studies (Chambers et al, 2003; Mitsui et al, 2003), *Nanog* deletion led to upregulation of PrE marker genes *Gata6*, *Gata4*, and *Sox17*, alongside downregulation of *Pax6* and *T*, markers of neural ectoderm and mesoderm, respectively (Fig. 1A). *Nanog* knockout resulted in notable morphological alterations (Fig. 1B). The colonies became smaller and exhibited characteristics indicative of differentiation, in contrast to the undifferentiated morphology of control RCNβH ESCs (Fig. 1B). These results indicate that *Nanog* plays a critical role in repressing PrE differentiation. To confirm that these gene expression changes were directly attributable to NANOG loss, we generated a doxycycline-inducible exogenous *Nanog* ESC line, RCNβH(iN) (Fig. 1C,D). The upregulated expression of *Gata6*, *Gata4*, and *Sox17* observed after 4-OHT treatment was reversed upon induced *Nanog* expression with doxycycline (Fig. 1E). We further investigated whether NANOG's DNA binding ability is required for this regulation by creating an inducible NANOG mutant, NANOG:N51A, defective in DNA binding (Navarro et al, 2012; Appendix Fig. S1B,C). In RCNβH(iN51A) ESCs, 4-OHT-induced upregulation of PrE markers was not rescued by mutant NANOG expression (Appendix Fig. S1D), indicating that NANOG's DNA binding is essential for repressing these genes.

### Nanog represses Tbx3 expression in ESCs

To explore *Nanog*'s regulatory mechanisms, we performed RNA sequencing (RNA-seq) comparing RCNβH and *Nanog*$^{-/-}$ ESCs,

identifying 1,395 significantly upregulated and 1,307 downregulated genes (Fig. 1F; Dataset EV1). Among upregulated genes were several PrE markers including *Gata6*, *Gata4*, *Sox17*, *Sox7*, and *Pdgfra* (Fig. 1F). NANOG loss also dysregulated genes involved in cell death, proliferation, and pluripotency (Appendix Fig. S1E), consistent with impaired colony formation (Appendix Fig. S1F) and reduced proliferation (Appendix Fig. S1G) observed in *Nanog*$^{-/-}$ ESCs. Gene ontology (GO) analysis of differentially expressed genes further supports *Nanog*'s role in differentiation control (Appendix Fig. S1E). Deletion of *Nanog* did not affect core pluripotency factor *Oct4* expression at transcript or protein levels (Fig. 1G; Appendix Fig. S1H), in agreement with previous report (Navarro et al, 2012). In contrast, the expression of *Sox2*, *Klf2*, and *Klf4* was reduced, whereas *Klf5* levels remained unchanged in *Nanog*$^{-/-}$ ESCs. Notably, both mRNA and protein levels of *Tbx3* were significantly increased in *Nanog*$^{-/-}$ ESCs (Fig. 1F,G; Appendix Fig. S1H), suggesting that *Nanog* represses *Tbx3* expression. As a key pluripotency factor with known roles in promoting mesendo-derm differentiation (Lu et al, 2011; Waghray et al, 2015; Weidgang et al, 2013; Zhang et al, 2019), TBX3 represents a potential mediator of Nanog-dependent PrE regulation. To examine whether *Nanog* directly modulates *Tbx3*, we assessed *Tbx3* expression in inducible RCNβH(iN) ESCs. Elevated *Tbx3* expression induced by 4-OHT in RCNβH(iN) ESCs was reversed by doxycycline-induced *Nanog* expression (Fig. 1H), whereas overexpression of *Nanog* in WT ESCs did not alter *Tbx3* levels (Fig. 1H). The mutant NANOG:N51A failed to reverse the elevated *Tbx3* expression upon 4-OHT treatment in RCNβH(iN51A) ESCs (Appendix Fig. S1I), confirming the DNA binding-dependent regulation of *Tbx3* by NANOG.

### Nanog regulates gene expression via controlling enhancer and promoter activity

To understand *Nanog*'s repressive function on *Tbx3* and PrE marker genes, we analyzed NANOG ChIP-seq data (Zhang et al, 2019). Most NANOG binding sites were distal to transcription start sites (TSSs), located in intergenic regions and gene bodies (Appendix Fig. S2A). Overlapping NANOG binding with ESC chromatin states (Chronis et al, 2017) revealed predominant binding at active enhancers marked by H3K27ac, H3K4me1, and H3K4me2 (Fig. 2A, states 3 and 4), as well as at active promoters enriched in H3K4me3, H3K27ac, and H3K9ac (Fig. 2A, state 1).

The text in the left column preceding "Results":

NANOG and Polycomb Repressive Complex 2 (PRC2) to maintain pluripotency by repressing key PrE genes, including Gata6, Gata4, and Sox17, through distal enhancers. Loss of Nanog releases PRC2-mediated repression of Gata6, initiating its expression independently of Tbx3. Subsequently, upregulated Tbx3 interacts with GATA6 to amplify PrE gene expression, promoting differentiation. These findings reveal a NANOG-TBX3-GATA6 regulatory axis that coordinates ESC self-renewal and PrE lineage commitment.

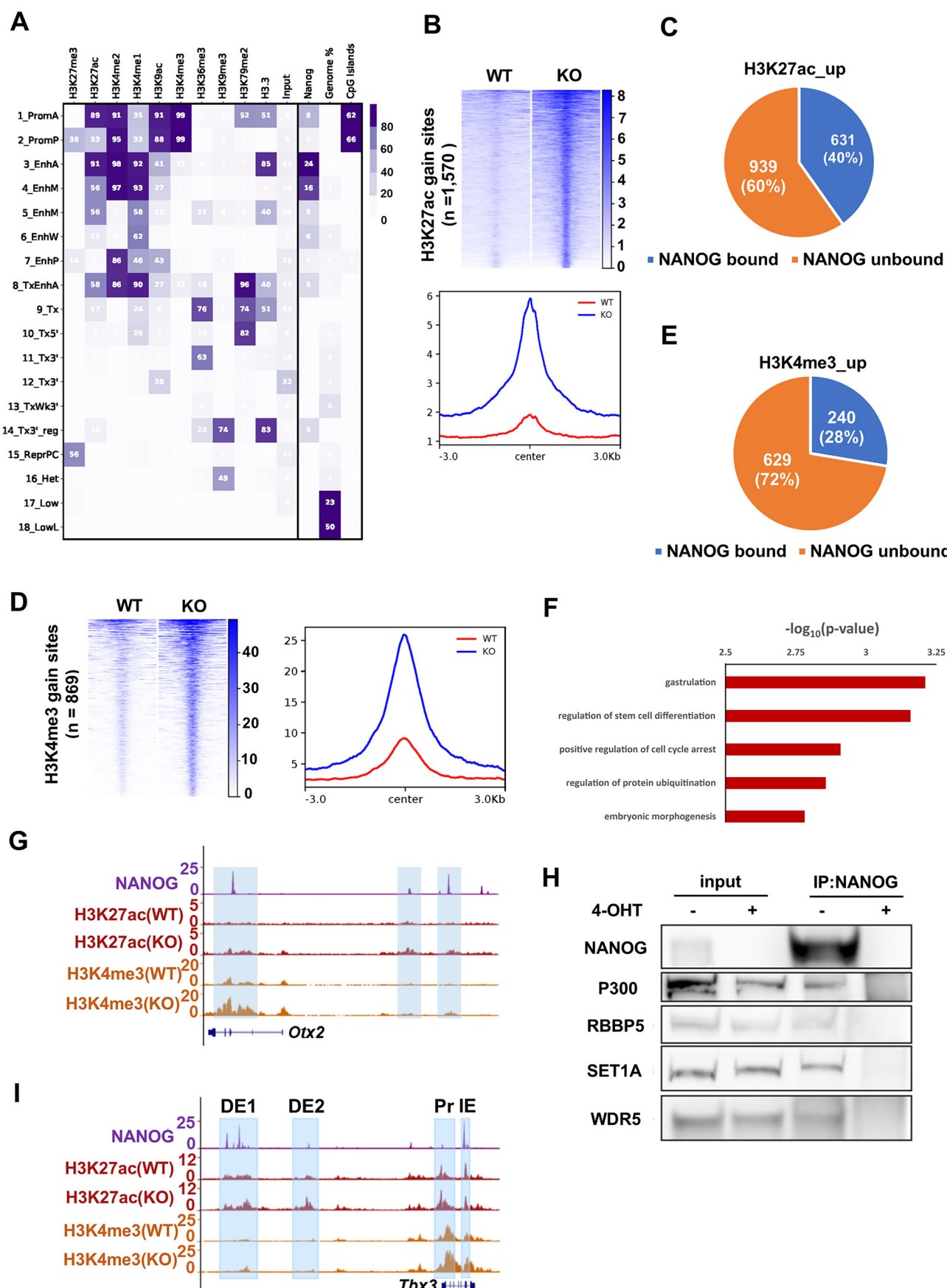

**Figure 2. *Nanog* regulates gene expression via controlling their enhancer and promoter activity.**

(A) Chromatin state enrichment of NANOG target sites in ESCs; ESC chromatin states were defined using ChromHMM (Chronis et al, 2017). Rows represent chromatin states and their mnemonics. Columns give the frequency of the indicated histone marks and H3.3 for each chromatin state (ChromHMM emission probabilities), color-coded from purple (highest) to white (lowest). Enrichment of NANOG in each chromatin state is shown. (B) Heatmap representation of normalized tag density profiles depicting increased H3K27ac in *Nanog* KO ESCs compared to RCNβH (WT) ESCs, along with corresponding metaplots illustrating signal intensities. (C) Number of genomic sites with an increase in H3K27ac (≥twofold difference) between RCNβH (WT) and *Nanog* KO ESCs bound by NANOG in WT ESCs. (D) Heatmap of normalized tag density profiles depicting increased H3K4me3 in *Nanog* KO ESCs compared to RCNβH (WT) ESCs, along with corresponding metaplots of signal intensities. (E) Number of genomic sites with an increase in H3K4me3 (≥twofold difference) between WT and *Nanog* KO ESCs bound by NANOG in WT ESCs. (F) GO analysis for biological processes associated with an increase in both H3K27ac and H3K4me3 upon *Nanog* deletion in ESCs. GO enrichment was performed using Metascape, with statistical significance assessed by a hypergeometric test and Benjamini–Hochberg correction for multiple testing. (G) Genome browser view of ChIP-seq tracks for NANOG, H3K27ac and H3K4me3 at the *Otx2* loci in RCNβH (WT) and RCNβHt (*Nanog* KO) ESCs. WT denotes RCNβH cells, KO represents *Nanog*⁻/⁻ cells. Highlighted in blue are regions that exhibit differential patterns between RCNβH and RCNβHt ESCs. (H) Confirmation of the interaction between NANOG and P300, RBBP5, SET1a, and WDR5 through Co-Immunoprecipitation (Co-IP) followed by western blot analysis. Lysates from cells were used for IP with anti-NANOG, while western blot analysis using anti-P300, anti-RBBP5, anti-SET1A and anti-WDR5. RCNβHt cells (*Nanog* KO) served as control. (I) Genome browser view of ChIP-seq tracks for NANOG, H3K27ac and H3K4me3 at the *Tbx3* locus in RCNβH (WT) and RCNβHt (*Nanog*⁻/⁻) ESCs. Highlighted in blue are region that exhibit differential patterns between RCNβH and RCNβHt ESCs. WT denotes RCNβH cells, KO represents *Nanog*⁻/⁻ cells, DE1 indicates distal enhancer 1, DE2 denotes distal enhancer 2, IE denotes internal enhancer (Zhang et al, 2019), and Pr refers to the promoter region. Source data are available online for this figure. Source data are available online for this figure.

NANOG also bound regions marked by H3K36me3, which promotes transcriptional elongation (Fig. 2A, states 5, 8, 14).

We next examined histone modifications H3K27ac and H3K4me3 in RCNβH and *Nanog*⁻/⁻ ESCs. While average H3K27ac signal at all NANOG binding sites was unchanged (Appendix Fig. S2B), 1570 sites showed ≥twofold increased H3K27ac upon *Nanog* deletion (Fig. 2B), and 3632 sites showed decreased H3K27ac (Appendix Fig. S2C). Notably, 40% of sites with increased and 65% with decreased H3K27ac were NANOG-bound in WT ESCs (Fig. 2C; Appendix Fig. S2D). Similarly, H3K4me3 showed no average global change (Appendix Fig. S2E), but 869 sites gained and 3949 sites lost H3K4me3 upon *Nanog* deletion (Fig. 2D; Appendix Fig. S2F). Among these, 28% of gained and 47% of lost sites were NANOG-bound (Fig. 2E; Appendix Fig. S2G). GO analysis linked increased H3K27ac and H3K4me3-sites to differentiation and cell cycle arrest genes (Fig. 2F), whereas decreased sites were linked to organ development, proliferation, and Wnt signaling (Appendix Fig. S2H). Examples include increased marks at the *Otx2* locus (Fig. 2G) and decreased marks at *Ccnd1* (Appendix Fig. S2I). These data suggest NANOG can function as both repressor and activator by modulating enhancer and promoter histone modifications. Supporting this, Co-immunoprecipitation showed physical interaction of NANOG with the histone acetyltransferase P300 and COMPASS complex components SET1A, RBBP5, and WDR5 (Fig. 2H), indicating NANOG influences enhancer/promoter activity through these chromatin modifiers.

## Nanog regulates Tbx3 expression via its promoter and enhancers

NANOG ChIP-seq revealed binding at both promoter and putative enhancers of the *Tbx3* gene (Fig. 2I), suggesting direct regulation. The enhancer DE1, essential for *Tbx3* expression (Zhang et al, 2019), showed increased H3K27ac upon *Nanog* deletion, as did another enhancer DE2 (Fig. 2I). H3K4me3 levels at the *Tbx3* promoter also increased in *Nanog*⁻/⁻ ESCs (Fig. 2I). A 2 kb *Tbx3* promoter reporter showed higher activity in *Nanog*⁻/⁻ ESCs, whereas a 1 kb promoter lacking the NANOG binding site was unaffected (Appendix Fig. S2J), confirming *Nanog* represses *Tbx3* via its promoter. Using dCas9 with guide RNAs targeting the

NANOG binding region at the *Tbx3* promoter reduced NANOG occupancy (Appendix Fig. S2K) and increased *Tbx3* expression (Appendix Fig. S2L), further confirming direct repression by NANOG at the promoter. Given NANOG's interaction with P300, and COMPASS components SET1A, RBBP5 and WDR5 (Fig. 2H), these results suggest *Nanog* represses *Tbx3* expression by modulating enhancer and promoter chromatin states.

## Nanog and Tbx3 collaboratively maintain ESC self-renewal by repressing PrE differentiation

*Tbx3* was among the most significantly upregulated pluripotency genes following *Nanog* deletion (Fig. 1F; Appendix Fig. S1H). Given *Tbx3*'s reported role in meso-endoderm differentiation (Lu et al, 2011; Waghray et al, 2015; Weidgang et al, 2013; Zhang et al, 2019), we hypothesized *Nanog* may regulate PrE differentiation partly through *Tbx3*. We generated *Tbx3* knockout ESCs by targeting exon 2 to induce a frameshift mutation (Appendix Fig. S3A,B). The expression of major pluripotency genes *Oct4, Sox2, Nanog, Klf5* remained largely unchanged in *Tbx3*⁻/⁻ ESCs (Appendix Fig. S3C), consistent with prior findings that *Tbx3* is dispensable for ESC maintenance (Russell et al, 2015). However, in *Nanog/Tbx3* double knockout ESCs, pluripotency gene expression was significantly reduced compared to either single knockout (Appendix Fig. S3C). Colony morphology and alkaline phosphatase staining also revealed significant differentiation in double knockout cells (Appendix Fig. S3D,E). These data indicate that *Nanog* and *Tbx3* cooperate to maintain ESC self-renewal. Contrary to previous RNAi studies reporting decreased endoderm markers upon *Tbx3* knockdown (Lu et al, 2011), we observed increased *Gata6, Gata4*, and *Sox17* expression in *Tbx3*⁻/⁻ ESCs (Fig. 3A). To exclude clonal variation, we generated a doxycycline-inducible *Tbx3* rescue line in *Tbx3*⁻/⁻ ESCs (RCNβH- *Tbx3*⁻/⁻ (iT)) (Appendix Fig. S3F). Induction of *Tbx3* reduced PrE marker expression compared to uninduced cells (Fig. 3B), confirming *Tbx3* represses PrE differentiation in ESCs.

To further explore the cooperative function of *Tbx3* and *Nanog* to maintain ESC self-renewal, we performed ChIP-seq of FLAG-tagged endogenous TBX3 in ESCs (Appendix Fig. S3G). TBX3 binding occurred primarily at transcription start sites (TSSs; 26.8%) and ±2 kb regions around TSSs (11.4%) (Fig. 3C). Overlapping

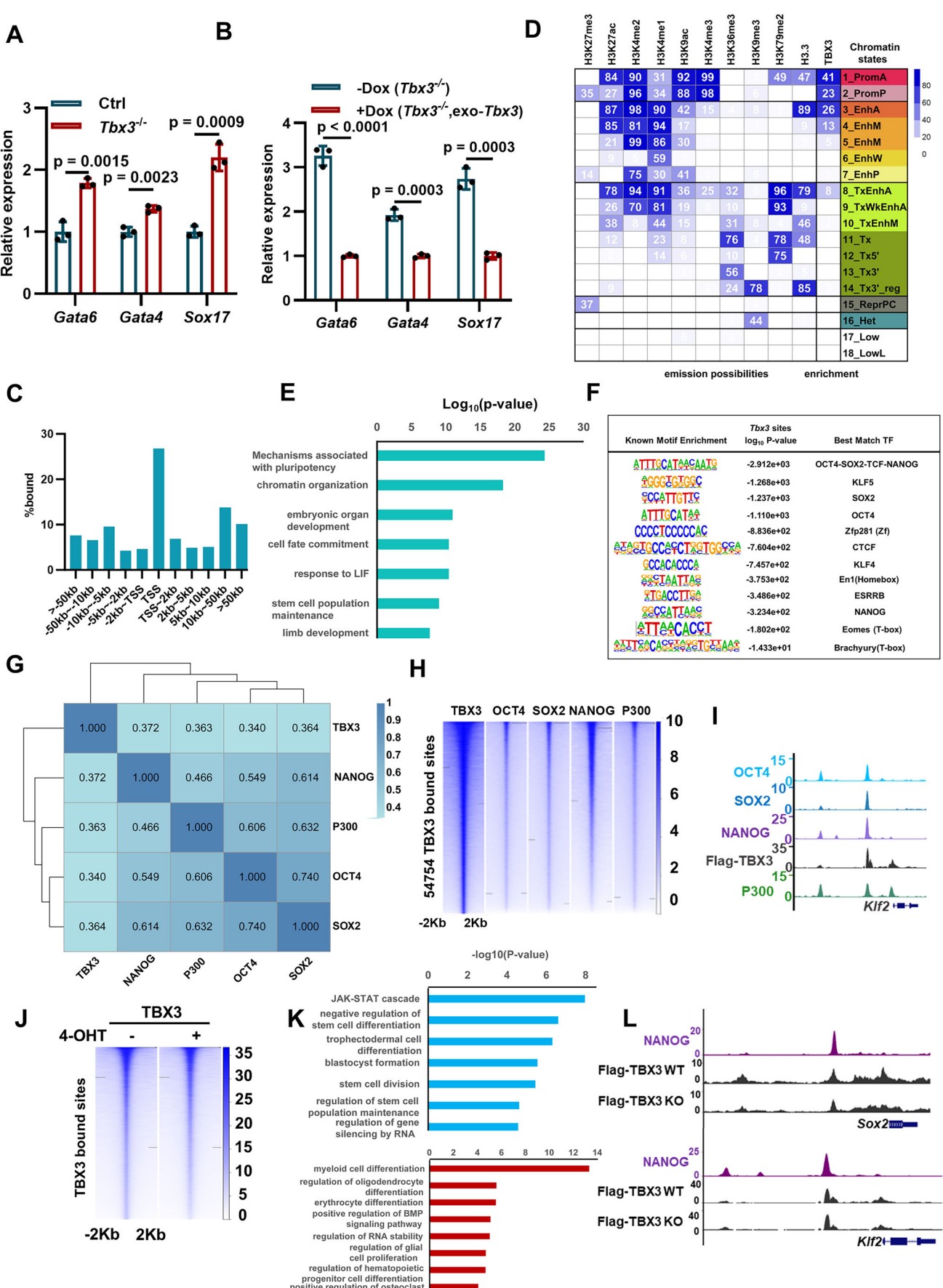

**Figure 3.   *Nanog* and *Tbx3* maintain the self-renewal of ESCs upon repressing PrE gene expression.**

(A) qPCR analysis for transcript levels of *Gata6*, *Gata4*, and *Sox17* genes in RCNβH (Ctrl.) and RCNβH-Tbx3$^{-/-}$ (*Tbx3*$^{-/-}$) ESCs. Data are presented as mean ± SD ($n = 3$, biological replicates). (B) qPCR analysis for transcript levels of *Gata6*, *Gata4*, and *Sox17* genes in RCNβH-Tbx3$^{-/-}$ (iT) ESCs with or without the addition of 1 μg/ml of Doxycycline for 4 days. Data are presented as mean ± SD ($n = 3$, biological replicates) *Gata6*, Row 1, $P = 6.03 \times 10^{-5}$. (C) Distribution of TBX3 target sites determined by ChIP-seq in ESCs in relation to their distance to TSSs. (D) Chromatin state enrichment of TBX3 target sites in ESCs. ESC chromatin states were defined in Chronis et al (Chronis et al, 2017) using ChromHMM. Rows represent chromatin states and their mnemonics. Columns give the frequency of the indicated histone marks and H3.3 for each chromatin state (ChromHMM emission probabilities), color-coded from blue (highest) to white (lowest). Enrichment of TBX3 in each chromatin state is shown in the last column. (E) Significant GO terms for genes with TBX3 target sites within 10 kb of their TSS. GO enrichment was performed using Metascape, with statistical significance assessed by a hypergeometric test and Benjamini–Hochberg correction for multiple testing. (F) Motifs identified at TBX3-bound sites by known motif search and the best matching TFs. (G) Hierarchical clustering of pairwise enrichments of TBX3, OCT4, SOX2, NANOG, and P300 binding sites in ESCs. (H) Heatmaps of normalized ChIP-seq signal for TBX3, OCT4, SOX2, NANOG, and P300 at all sites bound by TBX3 in ESCs. (I) Genome browser view of OCT4, SOX2, NANOG, TBX3, and P300 binding at the *Klf2* locus in ESCs. (J) Heatmaps of normalized ChIP-seq signal for TBX3 in RCNβH and *Nanog*$^{-/-}$ ESCs at TBX3 occupied sites in RCNβH ESC lines. (K) GO analysis for biological processes enriched in genes associated with the decreased (upper) and increased (down) TBX3 peaks upon *Nanog* deletion in ESCs. GO enrichment was performed using Metascape, with statistical significance assessed by a hypergeometric test and Benjamini–Hochberg correction for multiple testing. (L) Genome browser view of ChIP-seq tracks for NANOG and TBX3 at the *Sox2* (upper) and *Klf2*(down) loci in RCNβH (WT) and RCNβHt (*Nanog* KO) ESCs. WT denotes RCNβH cells, KO represents *Nanog*$^{-/-}$ cells. The unpaired two-tailed Student's *t* test was used for the statistical analysis in (A, B). Source data are available online for this figure. Source data are available online for this figure.

TBX3 binding sites with annotated ESC chromatin states (Chronis et al, 2017) revealed preferential TBX3 occupancy at active promoters and enhancers, marked by H3K27ac, H3K4me2, H3K9ac, and H3K4me3 (Fig. 3D). GO analysis associated TBX3-bound loci with pluripotency, cell fate commitment, LIF response, and stem cell maintenance functions (Fig. 3E), consistent with *Tbx3* roles in ESC self-renewal and differentiation (Esmailpour and Huang, 2012; Ivanova et al, 2006; Lu et al, 2011; Russell et al, 2015; Weidgang et al, 2013; Zhang et al, 2019; Zhao et al, 2014). Moreover, enriched motifs at TBX3 sites included KLF4, ESRRB, CTCF, OCT4, NANOG, SOX2, and the OCT-SOX-TCF-Nanog composite motif (Fig. 3F), highlighting TBX3's co-occupancy with core pluripotency factors, supported by genome-wide binding comparisons (Fig. 3G,H) and the *Klf2* locus example (Fig. 3I). Homer motif enrichment analysis also identified Zfp281, En1, and several TF-binding motifs, including the Eomes and Brachyury T-box binding motifs (Fig. 3F), which are consistent with the motifs observed in TBX3 ChIP-seq data from limb bud tissue (Soussi et al, 2024).

Upon *Nanog* deletion, TBX3 maintained a largely similar genome-wide binding profile (Fig. 3J), with 3599 peaks showing decreased and 2542 peaks showing increased binding (Appendix Fig. S3H,I). GO analyses indicated that genes near decreased TBX3 binding were linked to stem cell maintenance and differentiation, whereas genes near increased peaks were related to BMP signaling and differentiation of myeloid, oligodendrocyte, and other lineages (Fig. 3K). Co-localization of NANOG and TBX3 was detected at major pluripotency genes such as *Sox2* and *Klf2*, and the deletion of *Nanog* led to the reduction of TBX3 binding to them (Fig. 3L). In WT ESCs, Co-immunoprecipitation (Co-IP) showed the physical interaction between NANOG and TBX3 (Fig. 4A). These data suggest *Nanog* influences ESC self-renewal and differentiation partly by modulating TBX3 binding.

## Nanog and PRC2 cooperate to repress PrE differentiation

The Polycomb Repressive Complex 2 (PRC2) represses endodermal genes in ESCs (Boyer et al, 2006; Zhang et al, 2019). PRC2, along with the BAF complex, modulates mesendoderm differentiation partly via *Tbx3* regulation (Zhang et al, 2019). Co-IP showed physical interaction of NANOG with PRC2 components EED and

SUZ12 (Fig. 4A), suggesting cooperation in gene regulation. Chromatin profiling revealed 354 loci with increased H3K27me3 and 873 loci with decreased H3K27me3 upon *Nanog* deletion (Fig. 4B). NANOG loss significantly reduced the occupancy of PRC2 core components EED, EZH2, and SUZ12, as well as H3K27me3 levels at the *Gata6*, *Gata4*, and *Sox17* loci (Fig. 4C; Appendix Fig. S4A,B). This indicates that *Nanog* promotes PRC2-mediated repression of PrE genes.

To further investigate NANOG- and PRC2-mediated repression of PrE genes, mAID-Nanog ESCs were used for rapid auxin-inducible degron (AID)-mediated NANOG depletion (Appendix Fig. S4C; Li et al, 2022). *Gata6* expression was rapidly induced following IAA-mediated NANOG degradation in mAID-Nanog ESCs, as well as after 24 h of 4-OHT–induced *Nanog* deletion in RCNβH ESCs (Appendix Fig. S4D,E). In contrast, *Gata4* and *Sox17* were significantly upregulated only after prolonged NANOG depletion (72–96 h) or 4-OHT treatment (Appendix Fig. S4D,E). *Tbx3* expression was also increased upon IAA-induced NANOG depletion in mAID-Nanog ESCs (Appendix Fig. S4F,G). These results indicate that *Nanog* similarly regulates PrE genes and *Tbx3* in both mAID-Nanog and RCNβH ESCs.

*Gata6* expression increased significantly within 2 h of IAA-induced NANOG depletion in mAID-Nanog ESCs, whereas *Gata4* and *Sox17* remained largely unchanged even after 2 days of induction (Fig. 4D). Correspondingly, H3K27ac at the *Gata6* enhancer increased after 4 h of NANOG depletion (Appendix Fig. S4H), while H3K27me3 levels decreased within 1 h (Fig. 4E). Early histone modifications at *Gata4* and *Sox17* were minimal (Fig. 4E; Appendix Fig. S4H), suggesting that NANOG loss initially derepresses *Gata6*, which may subsequently regulate downstream PrE genes, consistent with Polycomb-mediated PrE priming in ESCs (Illingworth et al, 2016). Treatment of WT ESCs with the PRC2 inhibitor GSK126 elevated expression of *Gata6*, *Gata4*, and *Sox17* (Fig. 4F). In mAID-Nanog ESCs, 4-h IAA-induced NANOG depletion significantly upregulated *Gata6* but not *Gata4* or *Sox17*. Combined PRC2 inhibition and NANOG depletion produced additive upregulation of PrE genes (Fig. 4F), indicating that NANOG and PRC2 cooperate to repress PrE differentiation. In summary, in WT ESCs, NANOG physically interacts with TBX3 and PRC2 subunits (Fig. 4A), and they co-occupy the *Gata6*, *Gata4*, and *Sox17* loci (Fig. 4C; Appendix Fig. S4A,B). Deletion of *Nanog*,

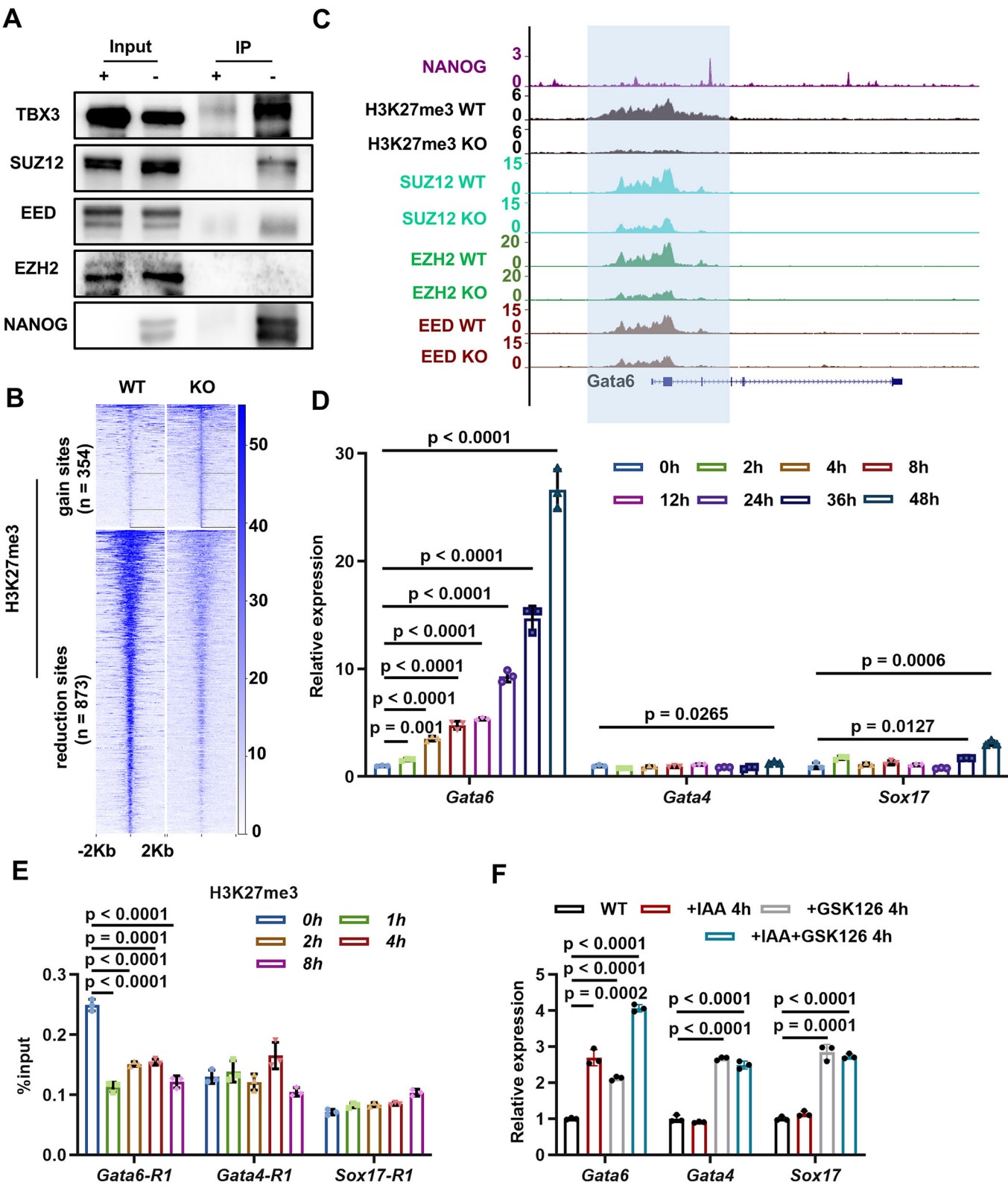

**Figure 4. Nanog facilitates PRC2 recruitment to repress primitive endoderm genes.**

(A) Confirmation of the interaction between NANOG and TBX3, SUZ12, EZH2 and EED in ESCs through Co-Immunoprecipitation (Co-IP) followed by western blot analysis. (B) Heatmap representation of normalized tag density profiles depicting increased and decreased H3K27me3 in WT and Nanog KO ESCs. (C) Genome browser view of ChIP-seq tracks of SUZ12, EZH2, EED binding as well as H3K27me3 in RCNβH (WT) and *Nanog* KO ESCs at the *Gata6* locus. WT denotes RCNβH cells, KO represents *Nanog*$^{-/-}$ cells. The promoter region is highlighted. The values on the *y* axis represent fold enrichment over control. (D) qPCR analysis of transcript levels for the *Gata6*, *Gata4*, and *Sox17* genes in Nanog-mAID ESCs treated with indole-3-acetic acid (IAA) for the indicated duration. Data are presented as mean ± SD ($n = 3$, biological replicates) *Gata6*, $P = 1.86 \times 10^{-5}$ (Row 1), $P = 3.28 \times 10^{-5}$ (Row 2), $P = 1.15 \times 10^{-5}$ (Row 3), $P = 5.39 \times 10^{-7}$ (Row 4), $P = 6.66 \times 10^{-5}$ (Row 5), $P = 4.74 \times 10^{-5}$ (Row 6). (E) H3K27me3 levels at the promoter regions of *Gata6*, *Gata4*, and *Sox17* genes in Nanog-mAID ESCs treated with IAA for the indicated duration, determined by ChIP-qPCR. Data are presented as mean ± SD ($n = 3$, biological replicates) *Gata6-R1*, $P = 8.79 \times 10^{-5}$ (Row 1), $P = 8.27 \times 10^{-5}$ (Row 3), $P = 4.59 \times 10^{-5}$ (Row 4). (F) qPCR analysis of relative mRNA expression levels of *Gata6*, *Gata4*, and *Sox17* in Nanog-mAID ESCs following 4-h treatments. Experimental groups included untreated control (WT), indole-3-acetic acid treatment ( + IAA), GSK126 treatment ( + GSK), and combined treatment with IAA and GSK126 ( + IAA + GSK). Data are presented as mean ± SD ($n = 3$, biological replicates) *Gata6*, $P = 8.32 \times 10^{-7}$ (Row 1), $P = 5.77 \times 10^{-6}$ (Row 2); *Gata4*, $P = 5.94 \times 10^{-5}$ (Row 1), $P = 1.11 \times 10^{-5}$ (Row 2); *Sox17*, $P = 5.02 \times 10^{-6}$ (Row 1). The unpaired two-tailed Student's *t* test was used for the statistical analysis in (D–F). Source data are available online for this figure. Source data are available online for this figure.

*Tbx3*, or PRC2 leads to derepression of these PrE genes (Figs. 1A,3A, B and 4F). These results indicate that NANOG, TBX3, and PRC2 act cooperatively to maintain ESC self-renewal by repressing PrE gene expression.

## Nanog regulates *Gata6*, *Gata4*, and *Sox17* expression via control of their distal enhancers and the access of TBX3

Contrasting the inhibitory effect of *Tbx3* on PrE genes in ESCs (Fig. 3A,B), previous studies have reported *Tbx3* as a promoter of meso-endoderm differentiation (Lu et al, 2011; Waghray et al, 2015; Weidgang et al, 2013; Zhang et al, 2019). Consistently, *Tbx3* deletion in *Nanog*$^{-/-}$ ESCs reduced expression of *Gata6*, *Gata4*, and *Sox17*, which was rescued by ectopic *Tbx3* expression (Fig. 5A), demonstrating *Tbx3*'s positive role in driving these PrE genes in the absence of *Nanog*. *Nanog* deletion led to the elevated *Tbx3* expression (Fig. 1E,F; Appendix Fig. S1H). We hypothesized that *Nanog* deletion increases TBX3 accessibility to regulatory elements of *Gata6*, *Gata4*, and *Sox17*, thus enhancing their expression.

Indeed, FLAG ChIP-seq revealed increased TBX3 binding at a distal enhancer (DE) ~187 kb upstream of the *Gata6* TSS upon *Nanog* deletion (Fig. 5B), in contrast to the decreased TBX3 binding at pluripotency genes (Fig. 3L). NANOG itself binds this region, and its loss led to elevated P300 occupancy and H3K27ac levels at the DE, implying *Nanog* suppresses *Gata6* expression by controlling DE activity. Physical interaction between NANOG and P300 further supports a model where *Nanog* competitively inhibits P300 binding at this enhancer (Fig. 2H). Targeted activation of this DE using dCas9-VPR with specific gRNAs in ESCs significantly upregulated *Gata6* but not *Gata4* or *Sox17* (Fig. 5C). Deletion of the *Gata6* DE reduced *Gata6* upregulation following *Nanog* deletion (Fig. 5D), indicating the DE's necessity for this effect. Interestingly, DE deletion also decreased elevated *Gata4* and *Sox17* expression in *Nanog*$^{-/-}$ cells (Fig. 5D), consistent with *Gata6*'s role in regulating these genes (Fujikura et al, 2002). Importantly, DE deletion did not completely abolish *Gata6* upregulation upon *Nanog* removal, indicating that additional enhancers contribute to NANOG-mediated repression of *Gata6* in ESCs. Consistent with this interpretation, multiple putative enhancer regions across the *Gata6* locus exhibited increased P300 occupancy and H3K27ac enrichment following *Nanog* deletion (Fig. 5B).

Similarly, TBX3 binding and P300 occupancy increased at enhancers ~93 kb upstream of *Gata4* and ~75 kb upstream of *Sox17*

TSSs in *Nanog*$^{-/-}$ ESCs (Appendix Fig. S5A,B). Activating these enhancers via dCas9-VPR increased *Gata4* or *Sox17* expression selectively, without affecting *Gata6* (Appendix Fig. S5C,D). Activating the enhancer of *Gata4* also increased *Sox17* expression (Appendix Fig. S5C), consistent with *Gata4*'s role in regulating *Sox17* (Holtzinger et al, 2010). Collectively, these results indicate that *Nanog* deletion enhances TBX3 binding and H3K27ac deposition at the enhancers of *Gata6*, *Gata4*, and *Sox17*, upregulating their expression. Functional validation showed that activating *Gata6* or *Gata4* enhancers promoted ESC differentiation toward the extraembryonic endoderm (ExE) lineage, as confirmed by morphological changes (Fig. 5E) and immunostaining for GATA6 and GATA4 (Fig. 5F; Appendix Fig. S5E).

## GATA6 collaborates with TBX3 to positively regulate PrE genes in *Nanog*$^{-/-}$ ESCs

Previous studies revealed that *Gata6* regulates *Gata4* and *Sox17* in WT ESCs (Fujikura et al, 2002). Knocking out *Gata6* in *Nanog*$^{-/-}$ ESCs markedly suppressed upregulation of *Gata4* and *Sox17* (Fig. 6A; Appendix Fig. S6A), and transient *Gata6* overexpression rescued their expression (Fig. 6B). ChIP-seq and ChIP-qPCR confirmed co-binding of GATA6 and TBX3 at *Gata4* and *Sox17* regulatory regions, with enhanced GATA6 occupancy in *Nanog*$^{-/-}$ ESCs (Appendix Figs. S5A,B and S6B). Conversely, *Gata4* knockout did not affect *Gata6* or *Sox17* expression after *Nanog* deletion (Appendix Fig. S6C–E), indicating *Gata6* is the primary regulator of these PrE genes in *Nanog*$^{-/-}$ ESCs. Self-regulation of *Gata6* was supported by increased GATA6 binding at its own promoter and enhancer following *Nanog* deletion (Fig. 6C), and by elevated endogenous *Gata6* expression upon exogenous GATA6 overexpression in *Nanog/Gata6* double KO ESCs (Fig. 6D).

*Tbx3* positively regulated *Gata6*, *Gata4*, and *Sox17* expression in *Nanog*$^{-/-}$ ESCs (Fig. 5A). *Gata6* expression was significantly increased at 2 h' IAA-induced NANOG degradation in mAID-Nanog ESCs (Fig. 4D), while *Tbx3* was significantly upregulated at 48 h' after NANOG degradation in mAID-Nanog ESCs (Appendix Fig. S4F,G). In RCNβH ESCs, *Tbx3* transcript and protein levels increased only after *Gata6* induction (Appendix Figs. S6F,G and S4E), suggesting a *Tbx3*-independent mechanism initiates *Gata6* expression. Indeed, *Nanog* deletion in *Tbx3* KO ESCs still induced *Gata6*, *Gata4*, and *Sox17* (Appendix Fig. S6H). These data support a model in which *Nanog* depletion induces *Gata6* expression

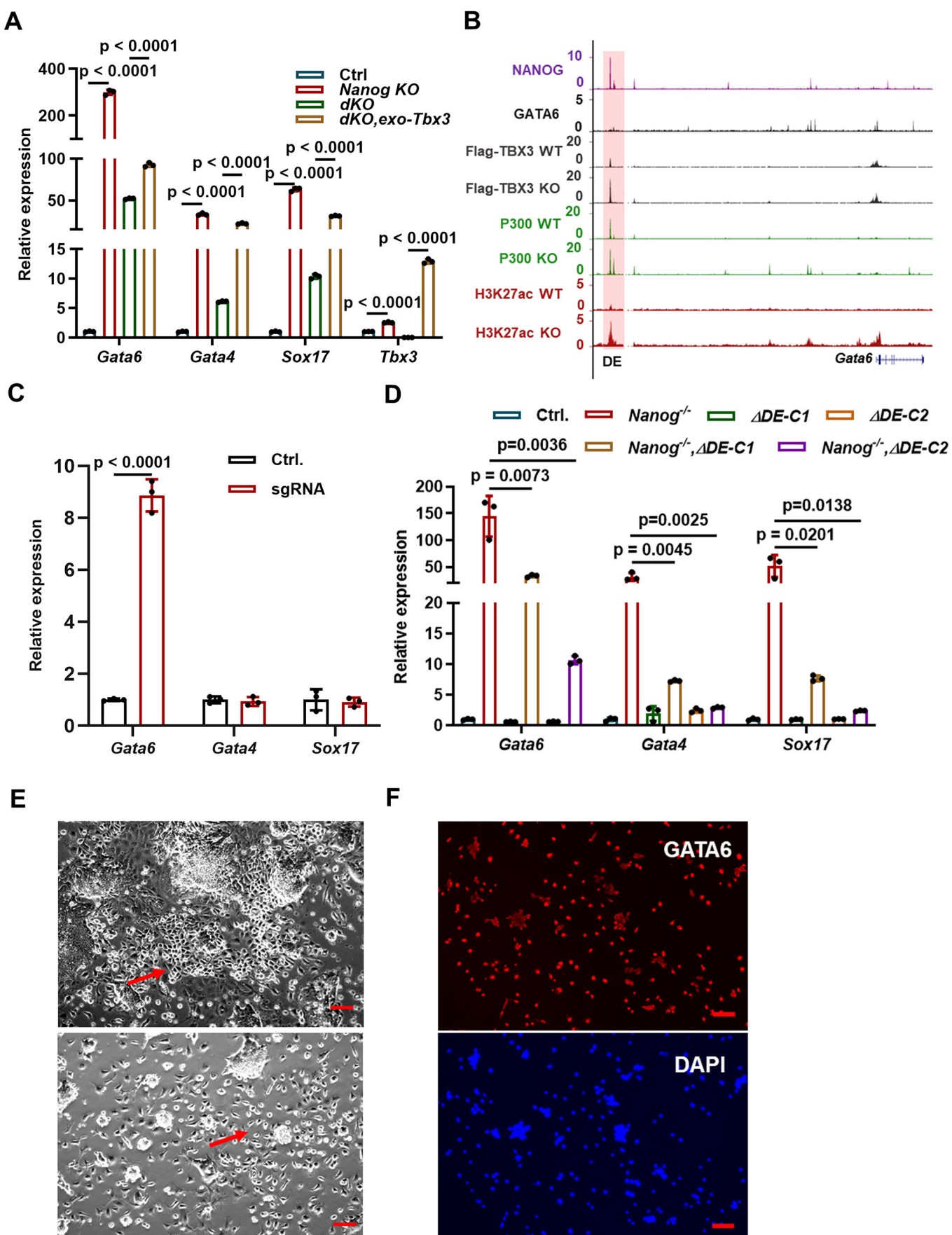

**Figure 5.  *Nanog* controls distal enhancers and TBX3 accessibility to regulate PrE genes.**

(**A**) qPCR analysis for transcript levels of *Gata6*, *Gata4*, *Sox17*, and *Tbx3* genes in RCNβH (Ctrl), RCNβH treated with 4-OHT (*Nanog*$^{-/-}$), RCNβH-Tbx3$^{-/-}$ (iT) ESc treated with 4-OHT (*Nanog/Tbx3* dKO), and RCNβH-Tbx3$^{-/-}$ (iT) ESCs treated with 4-OHT and Doxycycline (*Nanog/Tbx3* dKO, exo-*Tbx3*). Data are presented as mean ± SD (*n* = 3, biological replicates) *Gata6*, *P* = 1.36 × 10$^{-5}$ (Row 1), *P* = 1.12 × 10$^{-6}$ (Row 2); *Gata4*, *P* = 4.63 × 10$^{-6}$ (Row 1), *P* = 2.59 × 10$^{-6}$ (Row 2); *Sox17*, *P* = 1.39 × 10$^{-6}$ (Row 1), *P* = 1.07 × 10$^{-6}$ (Row 2); *Tbx3*, *P* = 3.99 × 10$^{-7}$ (Row 1), *P* = 1.66 × 10$^{-5}$ (Row 2). (**B**) Genome browser view of ChIP-seq tracks for NANOG and GATA6 in WT ESCs and TBX3, P300, and H3K27ac in RCNβH and RCNβHt (*Nanog* KO) ESCs at the *Gata6* locus. WT denotes RCNβH cells, KO represents *Nanog*$^{-/-}$ cells. The distal enhancer (DE) is highlighted. The values on the *y* axis represent fold enrichment over control. (**C**) qPCR analysis for transcript levels of *Gata6*, *Gata4*, and *Sox17* in ESCs co-transfected with deadCas9-VPR and gRNAs designed to target the DE region of *Gata6*. Data are presented as mean ± SD (*n* = 3, biological replicates) *Gata6*, *P* = 2.60 × 10$^{-5}$ (Row 1). (**D**) Transcript levels of *Gata6*, *Gata4* and *Sox17* in RCNβH (Ctrl) and RCNβHt (*Nanog*$^{-/-}$), and two DE KO ESC clones treated with and without 4-OHT based on qPCR. Data are presented as mean ± SD (*n* = 3, biological replicates). (**E**) Phase-contrast images depicting ESCs co-transfected with dCas9-VPR and gRNAs targeting the DE region of the *Gata6* gene (upper panel) or the *Gata4* gene (lower panel). Extraembryonic endoderm stem (XEN) cells are highlighted with arrows. Scale bars, 100 μm. (**F**) Immunostaining of GATA6 in ESCs co-transfected with deadCas9-VPR and gRNAs designed to target the DE region of *Gata6*. The cells were cultured and passaged in ESC medium without LIF for 4 to 5 times before immunostaining. Scale bars, 100 μm. The unpaired two-tailed Student's *t* test was used for the statistical analysis in (**A**, **C**, **D**). Source data are available online for this figure. Source data are available online for this figure.

independently of *Tbx3*. Mechanistically, this phase is accompanied by reduced H3K27me3 modification at the *Gata6* locus, relieving transcriptional repression (Fig. 4C,E). Consistently, *Nanog* depletion in *Tbx3*-knockout cells still leads to increased *Gata6* expression (Fig. 5A), albeit to a lesser extent than in cells expressing *Tbx3*. Together, these results indicate that the initial induction of GATA6 expression following *Nanog* depletion does not require *Tbx3*.

At a later stage, *Nanog* depletion results in gradual upregulation of *Tbx3* (Appendix Fig. S6F,G). Co-IP showed TBX3 interacts with GATA6, but not GATA4 (Fig. 6E), and both co-bind regulatory regions of PrE genes (Fig. 5B; Appendix Fig. S5A,B). Notably, *Nanog* knockout enhances Tbx3 occupancy at the DE enhancer of *Gata6* gene (Fig. 5B), establishing a positive feedback loop that further amplifies GATA6 expression. Deletion of *Tbx3* reduced GATA6 binding and reversed elevated H3K27ac and decreased H3K27me3 at *Gata6*, *Gata4*, and *Sox17* enhancers and promoters in *Nanog*$^{-/-}$ ESCs (Fig. 6F,G; Appendix Fig. S6I), confirming cooperative regulation by GATA6 and TBX3. Supporting this, in *Tbx3*-knockout cells, reintroduction of TBX3 followed by *Nanog* depletion yields higher *Gata6* expression than in knockout cells without TBX3 rescue (Fig. 5A), confirming that *Tbx3* synergizes with GATA6 to promote its expression during the late phase.

In summary, loss of *Nanog* upregulates *Tbx3* and *Gata6*, increasing their enhancer binding and promoting PrE differentiation. TBX3 interacts with GATA6 (Fig. 6E), and both co-bind *Gata6*, *Gata4*, and *Sox17* loci (Fig. 5B; Appendix Fig. S5A,B). Thus, GATA6 and TBX3 collaboratively regulate PrE gene expression.

### NANOG-TBX3-GATA6 regulatory network during early embryonic development

Single-cell RNA-seq (scRNA-seq) analysis of E3.5 and E4.5 mouse embryos (Nowotschin et al, 2019; Data ref: Nowotschin et al, 2019) revealed dynamic expression patterns consistent with this network. At E3.5, four Nanog-defined clusters included high (*Nanog*-H), medium (*Nanog*-M), and low (*Nanog*-L) *Nanog* populations (Appendix Fig. S6J). *Nanog*-H cells, representing uncommitted inner cell mass (ICM), co-expressed high pluripotency factor *Tbx3* and low *Gata6*, *Gata4*, and *Sox17* (Fig. 6H). In contrast, *Nanog*-L/ *Tbx3*-L clusters displayed elevated *Gata6*, *Gata4*, and *Sox17* with concomitant loss of pluripotency factors, marking progression toward PrE and indicating Tbx3-independent initiation of PrE gene

activation (Figs. 6H and 5A). At E4.5, distinct PrE and epiblast (EPI) clusters emerged (Appendix Fig. S6K), with *Tbx3* expression enriched in PrE, consistent with its role in promoting PrE genes alongside *Gata6* (Fig. 6I). Notably, *Nanog* and *Gata6* showed mutually exclusive expression in PrE and EPI (Fig. 6I), mirroring ESC observations.

Immunostaining further confirmed this dynamic: TBX3 and GATA6 were co-expressed at the 2-cell, 4-cell, 8-cell, and blastocyst stages (Appendix Fig. S6L), whereas by E4.5, TBX3 expression was confined to PrE cells (Appendix Fig. S6M), coinciding with its co-expression with PrE marker genes (Fig. 6I). Together, these analyses demonstrate that TBX3 operates in a dual mode supporting ESC self-renewal in concert with NANOG and promoting PrE differentiation in association with GATA6.

## Discussion

The reciprocal repression between NANOG and GATA6 is a central feature of lineage specification during pre-implantation development, directing cells of the inner cell mass (ICM) toward epiblast (Epi) or primitive endoderm (PrE) fates (Chambers et al, 2003; Chowdhary and Hadjantonakis, 2022; Frankenberg et al, 2011; Mitsui et al, 2003; Wamaitha et al, 2015). A reduction in NANOG levels has been shown to induce *Gata6* expression in the ICM (Frankenberg et al, 2011), and NANOG binding to the *Gata6* promoter has been reported to directly repress its transcription (Singh et al, 2007). Despite these observations, the molecular pathways by which NANOG suppresses PrE differentiation have remained incompletely defined. Our study provides new insight into this process by uncovering a regulatory mechanism in which NANOG cooperates with TBX3 and the PRC2 to repress PrE-associated genes through distal enhancer regulation. Loss of NANOG releases this repression, resulting in TBX3 upregulation and a functional switch whereby TBX3 partners with GATA6 to activate a PrE transcriptional program (Fig. 6J).

TBX3 has been previously described as a factor with dual functionality: maintaining ESC self-renewal under certain conditions while promoting differentiation toward mesodermal and endodermal lineages under others (Esmailpour and Huang, 2012; Ivanova et al, 2006; Lu et al, 2011; Russell et al, 2015; Weidgang et al, 2013; Zhang et al, 2019; Zhao et al, 2014). However, the molecular basis of this dichotomy has remained elusive. Our data

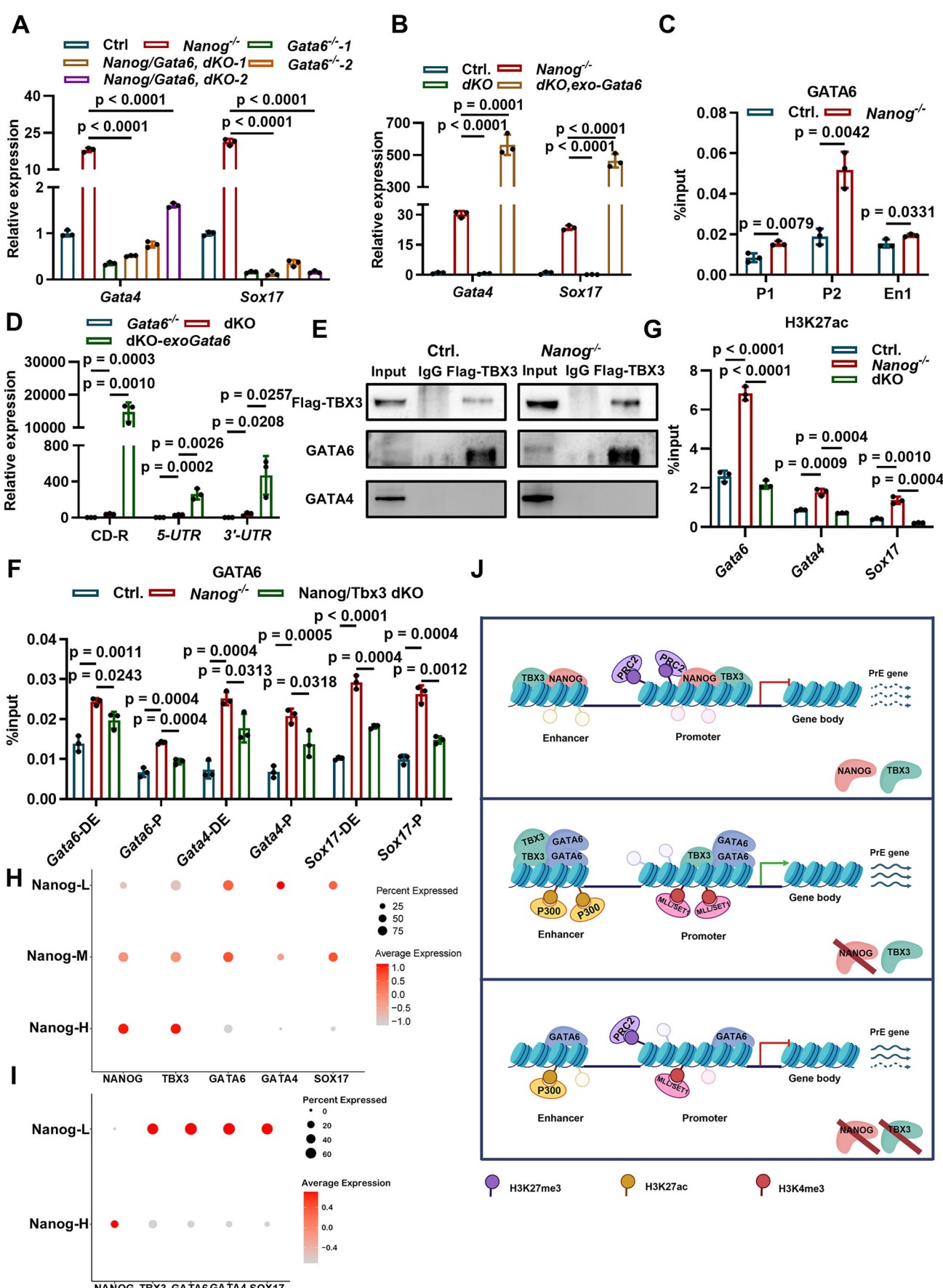

**Figure 6. TBX3 and GATA6 collaboratively regulate the expression of PrE genes *Gata6*, *Gata4* and *Sox17*.**

(A) qPCR analysis for transcript levels of the *Gata4* and *Sox17* in RCNβH (Ctrl), RCNβHt (*Nanog*$^{-/-}$), and RCNβH-Gata6$^{-/-}$ ESCs in the absence (*Gata6*$^{-/-}$), and presence (*Nanog/Gata6* dKO) of 4-OHT over a 4-day period. Data are presented as mean ± SD ($n = 3$, biological replicates) *Gata4*, $P = 5.49 \times 10^{-6}$ (Row 1), $P = 4.22 \times 10^{-6}$ (Row 2); *Sox17*, $P = 1.05 \times 10^{-5}$ (Row 1), $P = 1.05 \times 10^{-5}$ (Row 2). (B) qPCR analysis for transcript levels of the *Gata4* and *Sox17* in Ctrl (RCNβH), *Nanog*$^{-/-}$ (RCNβHt), *Nanog/Gata6* dKO, and *Nanog/Gata6* dKO with exo-Gata6 ESCs. Data are presented as mean ± SD ($n = 3$, biological replicates) *Gata4*, $P = 6.25 \times 10^{-6}$ (Row 2); *Sox17*, $P = 5.21 \times 10^{-5}$ (Row 1), $P = 3.90 \times 10^{-6}$ (Row 2). (C) GATA6 levels at *Gata6* promoter and enhancer regions in RCNβH (Ctrl.) and RCNβHt (*Nanog*$^{-/-}$) ESCs, determined by ChIP-qPCR. Data are presented as mean ± SD ($n = 3$, biological replicates). (D) qPCR analysis of transcript levels for *Gata6* in *Gata6*$^{-/-}$ (RCNβH-Gata6$^{-/-}$), *Nanog/Gata6* dKO (RCNβH-Gata6$^{-/-}$ t), and dKO, *exo-Gata6* (RCNβH-Gata6$^{-/-}$t with exogenous Gata6 expression) ESCs. Primers were designed based on coding, 5′-UTR, and 3′-UTR regions of the *Gata6* gene. Data are presented as mean ± SD ($n = 3$, biological replicates). (E) Confirmation of the interaction between Flag-TBX3 and GATA6 through Co-Immunoprecipitation (Co-IP) followed by western blot analysis. Lysates from RCNβH (Ctrl.) or RCNβHt (*Nanog*$^{-/-}$) cells were used for IP with anti-Flag, while western blot analysis using anti-GATA6 and anti-GATA4. Anti-IgG served as control. (F, G) ChIP-qPCR analysis of GATA6 (F) and H3K27ac (G) levels at the promoter and enhancer regions of the *Gata6*, *Gata4*, and *Sox17* genes in RCNβH (Ctrl), RCNβHt (*Nanog*$^{-/-}$), and RCNβH-Tbx3$^{-/-}$t (*Nanog/Tbx3* dKO) ESCs; Data are presented as mean ± SD ($n = 3$, biological replicates) (F) *Sox17*-DE, $P = 4.55 \times 10^{-5}$ (Row 1); (G) *Gata6*, $P = 8.07 \times 10^{-5}$ (Row 1), $P = 3.73 \times 10^{-5}$ (Row 2). (H) Dotplot showing indicated gene expression levels in *Nanog* low, middle and high clusters in E3.5 embryos. (I) Dotplot showing indicated gene expression levels in *Nanog* low and high clusters in E4.5 embryos. (J) A model for the regulation of PrE-differentiation by *Nanog*. The unpaired two-tailed Student's *t* test was used for the statistical analysis in (A–D, F, G). Source data are available online for this figure. Source data are available online for this figure.

suggest that the context-dependent role of TBX3 is dictated by its binding partners. In pluripotent ESCs, TBX3 engages with NANOG and PRC2 to silence PrE genes (Figs. 3–4; Appendix Fig. S3-S4), thereby stabilizing self-renewal. Both *Nanog* and *Tbx3* are core pluripotency regulators, and previous studies have shown that reduction of either gene induces ESC differentiation (Ivanova et al, 2006; Chambers et al, 2003; Mitsui et al, 2003). In agreement with these reports, we observed decreased expression of *Klf2* and *Klf4* upon *Nanog* deletion, and this reduction was further exacerbated when *Tbx3* was simultaneously deleted (Appendix Fig. S3C). Furthermore, ChIP-seq analysis revealed that NANOG and TBX3 co-occupy regulatory regions of key pluripotency genes, including *Oct4*, *Sox2*, *Klf4*, and *Klf2* (Fig. 3H,L). GO analysis of loci with reduced TBX3 binding in Nanog-deficient ESCs showed enrichment for biological processes associated with stem cell maintenance (Fig. 3K). Co-immunoprecipitation further confirmed the physical interaction between NANOG and TBX3 (Fig. 4A). Together, these observations support a cooperative role for NANOG and TBX3 in sustaining ESC self-renewal.

In contrast, upon *Nanog* deletion, PrE genes such as *Gata6* become strongly upregulated (Figs. 1A and 4D), coinciding with increased *Tbx3* expression and a pronounced enhancement of TBX3 binding at PrE gene enhancers (Fig. 5B). This shift in binding occurs in the absence of NANOG and is consistent with a "partner-switching" model, in which TBX3 transitions from cooperating with NANOG to maintain pluripotency to partnering with GATA6 to activate PrE gene expression. This model provides a mechanistic explanation for TBX3's dual activity and suggests a broader principle whereby pluripotency factors are repurposed to drive lineage specification once core repressors such as NANOG are lost. This partner-switching model not only reconciles previous observations of TBX3's dual activity but also establishes a broader principle in which pluripotency factors are redeployed to promote differentiation once core repressors are lost.

This mechanism parallels the context-dependent activity of ESRRB, which supports ESC identity in the presence of NANOG but drives PrE differentiation when NANOG is absent (Festuccia et al, 2016; Festuccia et al, 2018a; Festuccia et al, 2018b; Knudsen et al, 2023). Indeed, both TBX3 and ESRRB appear to safeguard pluripotency under normal conditions while assuming a differentiation-promoting role upon NANOG loss. Such functional plasticity may represent a general feature of the pluripotency

network, ensuring both the robustness of ESC maintenance and the capacity for rapid lineage commitment.

Our results further underscore the importance of *Gata6* autoregulation in enabling a swift and irreversible transition from pluripotency to PrE fate (Fig. 6A,B,D). We observed a rapid upregulation of *Gata6* within hours of *Nanog* depletion, followed by reinforcement of its own transcription and the activation of downstream PrE markers, including *Gata4* and *Sox17* (Fig. 4D; Appendix Fig. S4D,E). Autoregulatory loops are well-established mechanisms for generating bistability and irreversibility in developmental gene networks (Bai et al, 2007; Dasen et al, 1999; Leyva-Diaz and Hobert, 2019; Steinbach et al, 1998). In this study, autoregulation of *Gata6* may thus function as a feed-forward amplifier, stabilizing PrE differentiation once NANOG repression is lifted. Whether similar autoregulatory circuits operate in other PrE factors such as *Gata4* and *Sox17*, and whether their distinct dynamics contribute to lineage timing, remain open questions.

In summary, our findings reveal a regulatory axis in which NANOG, TBX3, and GATA6 integrate transcription factor switching, chromatin regulation, and autoregulatory mechanisms to balance self-renewal with lineage commitment. By providing mechanistic insight into how pluripotency factors are reconfigured during differentiation, this study offers a framework for understanding the plasticity of transcriptional networks that control early mammalian development.

## Methods

**Reagents and tools table**

| Reagent/resource | Reference or source | Identifier or catalog number |
| --- | --- | --- |
| **Experimental models** | | |
| RCNβH | Chambers et al, 2007 | N/A |
| RCNβH (iN) | This study | N/A |
| RCNβH (iN51A) | This study | N/A |
| RCNβH-Tbx3-/- | This study | N/A |
| RCNβH-Tbx3-/-(iT) | This study | N/A |
| RCNβH-ΔDE | This study | N/A |

| Reagent/resource | Reference or source | Identifier or catalog number |
|---|---|---|
| RCNβH-FlagTbx3 | This study | N/A |
| Nanog-mAID ESCs | Li et al, 2022 | N/A |
| RCNβH-Gata6-/- | This study | N/A |
| RCNβH-Gata4-/- | This study | N/A |
| **Antibodies** | | |
| Anti-Flag | Sigma | F1804 |
| Anti-OCT4 | Santa Cruz | SC5297 |
| Anti-SOX2 | Abcam | Ab97959 |
| Anti-β-ACTIN | Mesgen | Men1001 |
| Anti-RBBP5 | Cell Signaling Technology | Cat#13171 |
| Anti-SET1a | Abcam | ab70378 |
| Anti-EZH2 | Active Motif | Cat#39875 |
| Anti-EED | Millipore | CS204393 |
| Anti-TBX3 | Invitrogen | Cat#42-4800 |
| Anti-NANOG | Abcam | Ab80892 |
| Anti-SUZ12 | Abcam | Ab12073 |
| Anti-TUBULIN | Proteintech | 66240-1-Ig |
| Anti-GATA6 | R&D Systems | AF1700 |
| Anti-GATA4 | Santa Cruz | SC25310 |
| Anti-Rabbit IgG | Millipore | Cat#12370 |
| Anti-P300 | Active Motif | Cat#61401 |
| Anti-H3 | Proteintech | 17168-1-AP |
| Anti-H3K27me3 | Millipore | Cat#07449 |
| Anti-H3K27ac | Abcam | ab177178 |
| Anti-H3K4me3 | Abcam | ab8580 |
| Alexa Fluor® 594-Affini Pure Donkey Anti-Goat IgG | Jackson | Cat#705-585-003 |
| **Oligonucleotides and other sequence-based reagents** | | |
| PCR primers | This study | Dataset EV2 |
| qPCR primers | This study | Dataset EV2 |
| **Chemicals, enzymes and other reagents** | | |
| KnockOut™ DMEM | Thermo Fisher Scientific | Cat#10829018 |
| Penicillin/Streptomycin | Thermo Fisher Scientific | Cat#15140-122 |
| L-Glutamine | Thermo Fisher Scientific | Cat#25030081 |
| 2-Mercaptoethanol | Thermo Fisher Scientific | Cat#21985023 |
| ESGRO Recombinant Mouse LIF Protein | Sigma-Aldrich | ESG1106 |
| Lipo8000™ Transfection Reagent | Beyotime | C0533 |
| Trypsin-EDTA (0.25%), phenol red | Thermo Fisher Scientific | Cat#25200056 |
| Blasticidin | Invivogen | ant-bl-05 |
| Protease inhibitors | Roche | Cat#4693159001 |
| Paraformaldehyde | Sigma-Aldrich | P6148 |
| 1 M HEPES, pH 7.5 | Sigma-Aldrich | Cat#15630106 |
| IGEPAL CA-630 | Sigma-Aldrich | I8896 |

| Reagent/resource | Reference or source | Identifier or catalog number |
|---|---|---|
| Triton X-100 | Sigma-Aldrich | T8787 |
| 4-Hydroxytamoxifen | TOCRIS | Cat#3412 |
| Proteinase K | Thermo Fisher Scientific | Cat#25530049 |
| T4 DNA ligase | NEB | M0202S |
| DPBS | Vazyme | G101 |
| 0.5M-EDTA | Thermo Fisher Scientific | Cat#AM9262 |
| Pierce™ ChIP-grade Protein A/G Magnetic Beads | Thermo Fisher Scientific | Cat#26162 |
| HiScript II Q RT SuperMix | Vazyme | R223-01 |
| Taq Pro Universal SYBR qPCR Master Mix | Vazyme | Q712-02 |
| Next High-Fidelity 2× PCR Master Mix | NEB | M0541 |
| FBS | Gibco | Cat#10439024 |
| **Software** | | |
| FIJI – ImageJ | https://fiji.sc/ | |
| GraphPad Prism | https://www.graphpad.com | |
| Rstudio | https://www.rstudio.com/products/rstudio/ | |
| ChromHMM v1.1.0 | Chronis et al, 2017 | |
| Hisat2 v2.2.1 | Kim et al, 2019 | |
| MACS2 v2.2.7.1 | Zhang et al, 2008 | |
| BedTools v2.28.0 | Quinlan and Hall, 2010 | |
| Deeptools v3.5.1 | https://github.com/deeptools/deepTools | |
| HOMER v4.11.0 | http://homer.ucsd.edu/homer/ | |
| Sambamba v0.6.6 | https://github.com/biod/sambamba | |
| Fastp 0.20.0 | Chen et al, 2018 | |
| bigWigCorrelate | http://hgdownload.cse.ucsc.edu/admin/exe/ | |
| bedGraphToBigWig | http://hgdownload.cse.ucsc.edu/admin/exe/ | |
| **Other** | | |
| AP staining kit | Sigma-Aldrich | 86R-1KT |
| NovoRec® plus One step PCR Cloning Kit | Novoprotein | NR005-01B |
| MinElute PCR Purification Kit Qiagen | Agilent | Cat# 28004 |
| ECL Plus | Amersham | RPN2133 |
| Total RNA Isolation Kit | Hlingene | NG301S |
| BCA Protein Quantification Kit | Vazyme | E112-01 |

## Cell lines

Mouse ESC lines RCNβH, RCNβH (iN), RCNβH (iN51A), RCNβH-Tbx3$^{-/-}$, RCNβH-Tbx3$^{-/-}$(iT), RCNβH-ΔDE, RCNβH-

FlagTbx3, Nanog-mAID, RCNβH-Gata6$^{-/-}$, RCNβH-Gata4$^{-/-}$ were used in this study.

## Cell culture

ESCs were cultured in DMEM (Thermo Fisher) supplemented with 10% FBS, 1 mM sodium pyruvate, 2 mM L-glutamine, 0.1 mM 2-mercaptoethanol, and LIF on gelatin-coated plates.

## Colony formation assay

For colony formation experiments, approximately 1,000 cells were seeded in a 10-cm dish. After 8 days of culture, alkaline phosphatase (AP) staining was performed using the AP Staining Kit (Sigma). During this process, colony characteristics were categorized as follows: colonies with 90% or more AP-positive cells were labeled as undifferentiated, those with 5% or fewer AP-positive cells were classified as differentiated, and colonies with an intermediate percentage of AP-positive cells were designated as partially differentiated colonies.

## Plasmid construction

To generate sgRNA plasmids, PCR amplicons from pUC57 plasmid were ligated to Bsa1 digested pGL3-U6-2sgRNA-CCDB-EF1α-BSD vector. Correct plasmids were confirmed by Sanger sequencing. Oligo sequences were listed in Dataset EV2.

The cDNA of Nanog was cloned by PCR and ligated to the pPBH-TREtight-EGFP-TK-Hygro. Correct plasmids were confirmed by Sanger sequencing.

To generate the luciferase reporter construct, 2 kb promoter or 1 kb promoter region from *Tbx3* gene was amplified by PCR from mouse ES gDNA and ligated to the pGL3-basic vector. Correct plasmids were confirmed by Sanger sequencing.

## Generation of knockout ESC clones

To generate knockout cells, RCNβH cells were transfected with 1.5 μg of gRNA and 1 μg of Cas9 plasmids using Lipofectamine 8000 (Beyotime). After selection with 10 μg/ml blasticidin (BSD) (Invitrogen), colonies were picked up for genotyping and confirmed by Sanger sequencing.

## Generation of knock-in ESC clones

To generate knock-in cells, RCNβH cells were transfected with 1.5 μg of the targeting vector, 0.5 μg of gRNA and 1 μg of Cas9 plasmids using Lipofectamine 8000 (Beyotime). After selection with 10 μg/ml blasticidin (BSD) (Invitrogen), colonies were picked up for genotyping and confirmed by Sanger sequencing.

## Co-immunoprecipitation (Co-IP) and western blotting

ES cells were cultured in the absence and presence of 4-OHT addition over a period of 4 days. Then cells were lysed in lysis buffer (1% IGEPAL-CA-630, 50 mM Tris-HCL pH 8.0,150 mM NaCl, and protease inhibitors (Roche)). A fraction equivalent to 1/10th of the lysate was set aside as the input. Protein concentrations were quantified using the BCA Protein Assay Kit (Vazyme). For

immunoprecipitation, cell lysates were subjected to a 1-h incubation with the specified antibodies. Protein G beads (Thermo Fisher) were then added and incubated at 4 °C overnight. Following three washes with lysis buffer, the samples were mixed with 1×protein SDS loading buffer and boiled for 5 min. After boiling, the supernatant was allowed to cool on ice for 5 min before being loaded onto the gel for immunoblotting. Proteins were separated on a 4–12% Bis-Tris gel, transferred onto PVDF membranes, and subsequently probed with the appropriate antibodies. The blots were treated with secondary antibodies and visualized using ECL Plus (Amersham).

## Immunofluorescence staining

The cells were fixed with 4% paraformaldehyde for 10 min at room temperature. Subsequently, they underwent a blocking and permeabilization step with 3% serum in PBS containing 0.3% Triton X-100. Next, the cells were incubated with the indicated antibodies at 4 °C overnight. Following a series of washes, the cells were treated with Alex594-conjugated donkey anti-goat IgG (Jackson). Nuclei were visualized using DAPI staining.

## Dual-luciferase assays

ESCs were seeded in 24-well plates at an appropriate density for transfection. Co-transfection of the pGL3 basic vector, pGL3-1kb, or pGL3-2kb (each containing Tbx3 promoters of 1 kb or 2 kb length) along with pRL-TK plasmids encoding firefly and Renilla luciferase, respectively, was carried out using lipo8000 (Beyotime Biotechnology). Luciferase assay experiments were conducted using the Luciferase Assay System following the provided protocol (Promega). Initially, cells were washed twice with PBS and then lysed by the addition of Passive Lysis Buffer to each well, followed by shaking for 15 min at room temperature. Firefly luciferase activity was assessed by combining 20 μl of cell lysate with 100 μl of Luciferase Assay Reagent II (LAR II) and measuring luminescence. Subsequently, Renilla luciferase activity was measured by adding 20 μl of Stop & Glo Reagent to the same samples and recording luminescence again. The ratio of firefly to Renilla luciferase activity was determined for each sample to normalize firefly luciferase activity with respect to transfection efficiency.

## Quantitative RT-PCR

Cells were lysed using Total RNA Isolation Kit (Hlingene) for RNA exaction. Following this, cDNA was synthesized utilizing HiScript II Q RT SuperMix (Vazyme), and real-time PCR was performed using Taq Pro Universal SYBR qPCR Master Mix (Vazyme). Gene expression was quantified relative to *Gapdh* transcript levels, and standard deviation was computed from PCR triplicates. The error bars denote the standard deviation derived from three technical qPCR replicates in a representative experiment. The primer sequences utilized in the assay can be found in Dataset EV2.

## ChIP-seq

ChIP-seq was carried out following previously established protocols (Xu et al, 2020). In brief, ~$1 \times 10^7$ cells were fixed with 1% formaldehyde at room temperature for 10 min and then quenched with 0.125 M glycine. The cross-linked cells were resuspended in

sonication buffer and sonicated using a Bioruptor. Sonication comprised three rounds with low mode pulsing settings (30 s ON; 30 s OFF), followed by ten rounds with high mode pulsing settings (30 s ON; 30 s OFF). A one-thirtieth fraction of the sonicated chromatin was preserved as input. The remaining chromatin was incubated with 1 μg of antibody conjugated to magnetic beads overnight at 4 °C. After the immunoprecipitation, the beads were washed consecutively with RIPA buffer, low salt buffer, high salt buffer, LiCl buffer, and 1× TE buffer. The DNA library was generated utilizing Tn5. Subsequently, DNA was extracted by reversing crosslinking at 65 °C overnight with proteinase K (20 μg/mL) and then sequenced using the Illumina HiSeq 2500 platform.

## ChIP-qPCR

ChIP-qPCR was conducted following established procedures (Xu et al, 2020). In summary, ChIP assays were executed as previously mentioned, with the exception of the DNA library preparation step. Subsequent to the purification of immunoprecipitated DNA, 1 μL of the sample was allocated for each qPCR reaction. The qPCR was carried out using the Taq Pro Universal SYBR qPCR Master Mix (2×). Each qPCR was performed in duplicate, derived from a minimum of two independent experiments. The acquired data was then normalized to input values and calculated as the fold change relative to input. The ChIP-qPCR primers were designed based on the results of the ChIP-seq analysis and are detailed in Dataset EV2.

## RNA-seq

The extraction of total RNA was carried out using Total RNA Isolation Kit. Subsequently, sequencing was performed on Illumina HiSeq 2500 machines. The raw reads obtained were aligned to the reference genome (version mm10) employing hisat2. To generate the count table of genes, HT-Seq was employed with default parameters. Differential gene expression analysis was conducted using DESeq2 with its default parameters. The complete list of differential genes can be found in Dataset EV1.

## Quantification and statistical analysis

Statistical information for each experiment, including the total number of samples analyzed and the specific statistical tests, is presented as mean ± SD of three biological replicates unless otherwise stated. Statistical significance was determined by unpaired two-tailed Student's $t$ test. Exact $P$ values are indicated in the graph unless $P < 0.0001$.

## Data analysis

For ChIP-seq data, sequencing adapters were trimmed by fastp version 0.20.0 (Chen et al, 2018) with "-l 25 --detect_adapter_for_pe" flag. Reads were mapped to the mouse mm10 reference genome using HISAT2 version 2.2.1 (Kim et al, 2019) with "--no-temp-splicesite --no-spliced-alignment" option. Only the mapped reads with a mapping quality over 30 and properly paired reads were retained using samtools with "-f2 -F3844 -q 30" options. Duplicates were removed by sambamba version 0.6.6. The de-duplicated reads were utilized for peak calling with MACS2 version 2.2.7.1 (Zhang et al, 2008) with "-q 0.05 --keep-dup all -f BAMPE -g mm" options. In the

case of H3K36me3, the "--broad" option was additionally applied. In subsequent analysis, only peaks present in both replicates were considered using intersectBed from the bedtools suite (Quinlan and Hall, 2010). Motifs identification and calculation of distance to TSS were conducted using findMotifsGenome.pl and annotatePeaks.pl from the HOMER suite (Heinz et al, 2010) (version 4.11.0) on the narrowPeak files obtained from MACS2.

To analyze differential levels of histone modifications and differential binding by TBX3, the union peak set from the two genotypes was created by combining the narrowPeak files and merging peaks with a minimum overlap 1 bp using the mergeBed tool from the bedtools suite. Subsequently, the count of mapped reads across the union peaks from each condition was counted using coverageBed from the bedtools suite. Differential analysis was performed using the DESeq2 version 1.38.3 in R, considering genomic sites as differential peaks if there was at least a twofold difference observed between the wild-type and knockout samples.

In addition, to find the NANOG binding sites that overlap with regions showing differential levels of H3K27ac and H3K4me3, bedtools intersect was used. The distribution of H3K27ac and H3K4me3 near NANOG peaks in RCNβH and $Nanog^{-/-}$ cells was assessed using annotatePeaks.pl from the HOMER suite.

For RNA-seq data analysis, reads were aligned to the mouse genome assembly mm10.GRCm38 using HISAT2 version 2.2.1 using default parameters. Gene expression was quantified by featureCounts version 2.0.3. Differential gene expression analysis was performed via DESeq2 version 1.38.3 in R. The screening criteria for identifying differentially expressed genes were defined as Padj < 0.05 and |log2FoldChange| >0.5. Clustering and visualization of differentially expressed genes were conducted using pheatmap version 1.0.12 and ComplexHeatmap version 2.14.0. The complete list of differential genes can be found in Dataset EV1.

## GO analysis

Enriched biological processes were analyzed using Metascape (http://metascape.org) and GREAT (http://great.stanford.edu/public/html/) to identify significantly enriched terms ($P$ value < 0.01) within genes and genome binding regions.

## ChromHMM analysis

The ESCs chromatin state segmentations were obtained from the previous study conducted in mouse embryonic stem cells (Chronis et al, 2017). To assess the enrichment of NANOG and TBX3 binding events in distinct chromatin states, we utilized the ChromHMM OverlapEnrichment function and followed the instructions as previously described (Chronis et al, 2017). The dataset was binarized using the BinarizeBam function with default parameters. Then the LearnModel function identified 18 distinct chromatin states. NANOG and TBX3 binding events were then compared to these 18 established chromatin states using the OverlapEnrichment function from ChromHMM. The results were visualized using the pheatmap package in R.

## TF and histone modification enrichment analysis

ChIP-seq datasets for OCT4, SOX2, P300 and NANOG in mESCs were acquired from the prior study (Chronis et al, 2017). To create

the heatmaps in Fig. 3H,J, coverage profiles of mapped reads were generated using bamCoverage from deeptools with a normalization option --normalizeUsing RPKM. To compute TF enrichment relative to TBX3 sumits, comupteMatrix from deeptools was utilized, employing reference-point mode with parameters --referencePoint center -b 2000 -a 2000 -bs 100. Then we displayed the output matrix using plotHeatmap.

The distribution of H3K27ac and H3K4me3 near differential peaks in RCNβH and *Nanog*<sup></sup> $Nanog^{-/-}$ cells (Fig. 2B,D; Appendix Fig. S2C,F) was analyzed using comupteMatrix from deeptools and visualized with plotHeatmap and plotProfile.

### TF clustering and correlation analysis

Bedgraph files generated by MACS2 were converted to BigWig format with UCSC-bedGraphToBigWig (http://hgdownload.cse.ucsc.edu/admin/exe/), and the correlation analysis was performed using UCSC-bigWigCorrelate (http://hgdownload.cse.ucsc.edu/admin/exe/). The results were visualized using the pheatmap package in R.

### Single-cell RNA-Seq analysis

The scRNA-seq data from E3.5 and E4.5 mouse embryos were obtained from GSE123046 (Data ref: Nowotschin et al, 2019). Firstly, the R package Seurat version 4.3.0 was used to load the expression matrix and create data objects. The distribution of the number of detected genes (nFeature_RNA) for each cell were visualized using a violin plot. Then outliers were removed based on the plot. For the E3.5 data, cells from the first sample with log_nFeature_RNA < 7 were removed and cells from the second sample with log_nFeature_RNA < 6.5 were removed. For the two samples from E4.5, cells with log_nFeature_RNA < 6.5 were removed in both samples. Then cells with more than 20% of mitochondrial genes were also removed. The remaining number of cells was comparable to the number of the original publication. Then the merge function was used to combine the two samples of E3.5 and the two samples of E4.5, respectively. Data was normalized by using the NormalizeData function with a default scale parameter of 10,000. After data normalization, the highly variable genes were identified, and the top 2000 genes were used for downstream analysis. Before cell clustering, the harmony package version 1.2.0 was used to eliminate batch effects and then uniform manifold approximation and projection (UMAP) was used to visualize the data on a two-dimensional plot. The Jackstraw function was employed to determine the significant principal components for each sample. Specifically, 12 principal components were chosen as suitable parameter for cell clustering in E3.5 samples, while 15 principal components were selected for E4.5 samples. Then all cells were clustered by the FindClusters function from Seurat at a resolution of 0.5. These clusters were annotated based on cell type-specific markers.

## Data availability

The datasets produced in this study are available in the following databases: - RNA-seq and ChIP-seq data: NCBI Sequence Read Archive (SRA) PRJNA1043751.

The source data of this paper are collected in the following database record: biostudies:S-SCDT-10_1038-S44319-026-00707-6.

## Peer review information

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

## Acknowledgements

This work was supported by Ministry of Science and Technology 2022YFA1104300 (Z.W), and the National Natural Science Foundation of China (grants 32470842, 32170797, 31970812). The synopsis image and the image in Fig. 6J were created with BioRender.com.

## Author contributions

**Hao Wu**: Data curation; Investigation; Writing—original draft; Writing—review and editing. **Ying Ye**: Data curation; Investigation; Writing—original draft; Writing—review and editing. **Hongxia Dai**: Investigation. **Peixin Chen**: Investigation. **Tenghui Yang**: Investigation. **Zhifang Li**: Investigation. **Li Li**: Writing—review and editing. **Chirag Parsania**: Investigation. **Junjun Ding**: Supervision. **Man Zhang**: Supervision. **Erwei Zuo**: Supervision. **Ulf Schmitz**: Supervision; Investigation. **Xi Chen**: Supervision; Investigation. **Zhexin Zhu**: Investigation. **Wensheng Zhang**: Conceptualization; Supervision; Funding acquisition; Writing—original draft; Project administration; Writing—review and editing.

Source data underlying figure panels in this paper may have individual authorship assigned. Where available, figure panel/source data authorship is listed in the following database record: biostudies:S-SCDT-10_1038-S44319-026-00707-6.

## Disclosure and competing interests statement

The authors declare no competing interests.

