## [Peer Review File · EMBO Reports]

Nanog mediated control of TBX3-GATA6 circuitry in primitive endoderm differentiation of mESCs

Wensheng Zhang, Hao Wu, Ying Ye, Hongxia Dai, Peixin Chen, Tenghui Yang, Zhifang Li, Li Li, Chirag Parsania, Junjun Ding, Man Zhang, Erwei Zuo, Ulf Schmitz, Xi Chen, and Zhexin Zhu

Corresponding author(s): Wensheng Zhang (zhangwensheng@suda.edu.cn), Xi Chen (chenx9@sustech.edu.cn), Zhexin Zhu (zxzhu@ihm.ac.cn), Ying Ye (yeying@qlmu.edu.cn)

Review Timeline:

Submission Date:	20th Sep 24
Editorial Decision:	26th Nov 24
Appeal Received:	4th Dec 24
Editorial Decision:	23rd Dec 24
Appeal Received:	23rd May 25
Editorial Decision:	1st Aug 25
Appeal Received:	2nd Sep 25
Editorial Decision:	18th Sep 25
Revision Received:	18th Sep 25
Editorial Decision:	14th Nov 25
Revision Received:	1st Dec 25
Accepted:	14th Jan 26

Transaction Report:

25th Nov 2024

Dear Prof. Zhang,

Thank you for the submission of your research manuscript to EMBO reports. I sincerely apologize for the delay in handling your manuscript but we have now received the three enclosed reports on it.

I am sorry to say, that the evaluation of your manuscript is not a positive one. As you will see, while all referees acknowledge the potential interest of your findings to the more immediate field, they also raise important criticisms regarding the conclusiveness and technical quality of the study.

Given these concerns, the amount of work required to address them, and given the lack of enthusiastic support from the referees also regarding the broader conceptual advance provided, I am sorry to say that we cannot offer to publish your manuscript.

While we cannot pursue this manuscript further, we encourage you to transfer your study to our not-for-profit open-access sister journal, Life Science Alliance (LSA). We shared your manuscript and the accompanying reviews with LSA Executive Editor, Eric Sawey, who is interested in these findings, and would like to invite further consideration of this manuscript at LSA, revised to address the Reviewers' comments.

We understand that such a revision might need to be re-reviewed, in which case, Dr. Sawey will walk the Reviewers through our transfer process. We encourage you to use the link below to transfer your manuscript to LSA. You do not need to revise the manuscript before transferring it to LSA. Once you transfer, Dr. Sawey will email you an invitation to revise and resubmit, listing the same revision requests as mentioned above. Please feel free to reach out at e.sawey@life-science-alliance.org if you have any questions about the LSA journal, the transfer process or the revisions requested.

Yours sincerely,

=====

Referee #1:

In this manuscript, the authors reported a novel regulatory cue to direct differentiation of mouse embryonic stem cells (ESCs) into the extraembryonic endoderm (ExEn) cells. Using the inducible knockout ESCs, they revealed that the homeobox transcription factor Nanog represses the expression of the ExEn-specific transcription factors Gata6, Gata4 and Sox17. Nanog also represses the expression of pluripotency-associated transcription factor Tbx3. Interestingly, Tbx3 cooperate with Nanog to repress the ExEn-specific transcription factors. On the other hand, Tbx3 involves in the transcriptional activation of these genes in the absence of Nanog. These results suggested the dual role of Tbx3 in the differentiation to ExEn.

It was reported that the competitive function of Nanog and Gata6 involves in the lineage segregation of epiblast and ExEn. The finding shown in this report demonstrated the regulatory mechanism in more comprehensive manner by accompanying Tbx3. It is potentially worth to be published in EMBO Reports after revision for the points listed in below.

1. The authors use RCN β H ESCs for inducible knockout of Nanog in ESCs to show the induction of ExEn markers after elimination of Nanog. How about the morphological differentiation after elimination of Nanog? Does the the induction of ExEn markers depend on the culture condition i.e external signals? Wah happens in serum-free 2iLIF culture?
2. The phenotype of Tbx3 KO and Nanog: Tbx3 double KO ESCs is important part of this manuscript. Do they undergo morphological differentiation to ExEn? How about the self-renewal capacity in 2iLIF?
3. The role of PRC2 on ExEn differentiation has not been described in the references cited by the authors. It should be assessed in their ESC models with Ezh inhibitor.
4. Waghray et al (PMID:26095607) reported that the Tbx3 KO embryos possesses slightly higher numbers of ExEn cells at E4.5, suggesting that its function is dispensable for differentiation of ExEn. How is it consistent to the idea that Tbx3 facilitates ExEn differentiation shown by the authors?
5. In Fig 3H, the authors showed the result of enriched motifs at the Tbx3 binding sites. Why is the T-box motif missing? Doesn't it bind to specific DNA sequence directly? The ChIP-seq of Tbx3 in limb bud showed the enrichment showed the enrichment of Eomes (T-box) motif, suggesting the direct binding of Tbx3 to the specific DNA sequence.
6. Page 6: The authors stated that Nanog represses Gata6 expression by competitively inhibiting P300 binding to the DE region. What does it mean?
7. Page 11, middle: Zfp81 should be Zfp281.

Referee #2:

In this paper, Wu et al. define a gene regulatory network composed of Nanog, Tbx3 and Gata6, whose interplay modulates the balance between embryonic stem cell (ESC) self-renewal and primitive endoderm (PrE) differentiation. As Tbx3 is a pluripotency factor, the paper suggests it is able to both promote stem ESC self-renewal by suppressing expression of PrE markers and drive PrE differentiation, depending on its context. In particular, association of Tbx3 with Nanog and PRC2 stimulates the repression of PrE transcription. In absence of Nanog, Tbx3 is able to form a complex with Gata6, which in turn activates the expression of PrE-promoting genes.

While the a number of previous studies have highlighted a role for TBX3 in both pluripotency and differentiation, the mechanism by which it acts and the relevance of transcription stoichiometries has not been explored. For this reason, this paper would be suitable for EMBO reports following significant revision. However, the manuscript is both confusing and missing fundamental controls. At the level of mechanism, their model implies TBX3 and NANOG bind DNA together to recruit PRC2, but in Figure 3, they claim NANOG blocks TBX3 binding. Perhaps this could be addressed by titrating the level of these two factors and assessing their impact on differentiation. Moreover, they use a NANOG degron, but never show how IAA influences NANOG levels, but always show the influence of IAA indirectly via endodermal transcription or H3K27 modification. Finally, the writing in the manuscript is very confusing and it is unclear how to reconcile the different observations contained in the figures or what the final model is meant to be.

Specific points

- S11 it is unclear that the relative expression plotted is Tbx3, so adding plot title would help with clarity as they did in Fig. 1G.
- Figure S2D. In the text, this figure is used to make the point that the Nanog-bound sites that lose H3K27ac are predominantly in enhancers ("[...] 40% of sites with an increase in H3K27ac levels and 65% of sites with decreased in H3K27ac were bound by NANOG in WT ESCs (Figure 2C and Figure S2D) and located predominantly in enhancers (Figure S2D), indicating that NANOG contributes to the regulation of H3K27ac at active enhancers in ESCs."), but there is no explanation of how these enhancers were defined. Are these sites marked with H3K27ac and H3K4me3 (normally associated with promoters)? Or are these just sites that had H3K27ac in the WT but lose it in the Nanog KO? Why would then these be enhancers? Some clarification is needed.
- Fig.2I. Naming the tracks K27ac WT and KO could be misleading as the genotype actually refers to NANOG. I would consider changing the colors/transparency level to distinguish WT/KO or clarifying in a legend.
- Fig.S2K No explanation of what DE, R1 and R2 stand for is found in text or figure legend.
- Figure S3D The Tbx3 KO ESCs colonies look fine, but the double KO (Nanog/Tbx3) can't make colonies and seems to lose self renewal. If Nanog regulates Tbx3 in their GRN, why is the double mutant more severe?
- Figure 3B and S3D. The authors conclude that Tbx3 acts as a repressor of PrE differentiation based on the induction of Gata6, Gata4 and Sox17 upon Tbx3 KO. However, the AP staining in Fig.S3D actually shows that the Tbx3 KO cells are still able to generate stem cell colonies. This implies that it isn't required to suppress endoderm differentiation.
- Figure S3N, the increase in TBX3 binding upstream of the Sox17 and Gata4 loci is modest, is it statistically significant?
- Figure 4A, there is no figure showing downregulation of Nanog in mAID-Nanog ESCs upon IAA treatment, a quick Western Blot to mirror the relevant timepoints of this experiment might be useful. Furthermore, the authors suggest "The earlier initiation of Gata6 expression than both Gata4 and Sox17 suggest that Gata6 may facilitate the upregulation of Gata4 and Sox17 upon Nanog deletion (Figure 4A).", but even after 48 hours in absence of Nanog, the increase in Gata6 is never accompanied by significant levels of Gata4 and Sox17, why?
- Figure 4D, how much time in the absence of Nanog produced increased levels of endodermal transcription. It is very difficult to compare this to the degron lines. Were the degron lines a complete loss of function (see above)? How long are the conditional mutants cultured in the absence of Nanog.
- Fig 3N, shows a very modest GATA6 binding at its own DE. However, its binding elsewhere seems much more dramatic.
- Fig S4H shows Tbx3 levels in absence of Nanog remain relatively constant, but this contradicts the data in Figure 1. What do they think is the real result?
- Fig 4M shows high expression of Tbx3 in Nanog-high cells, based on reanalysis of in vivo single cell RNAseq. Based on this analysis, the levels of Nanog and Tbx3 appear correlated, but this would appear in contrast to the observation that TBX3 is expressed at high levels in Nanog mutants (Fig. 1).
- The manuscript makes multiple contradictory statements about the relationship between Nanog and Tbx3, stating the former represses the latter in the first paragraphs (or, in vitro), while claiming the opposite in the last section of results (in vivo). As a result it is very difficult to discern what the authors mean.
- Finally, it is not clear why the authors do not discuss the analogy to the model in Knudsen et al 2023, but simply refer to this as appearing during manuscript preparation as this paper has been in the public domain for sometime.

Referee #3:

In this manuscript by Wu and colleagues, the authors use mouse embryonic stem cells to investigate the potential mechanisms by which primitive endoderm differentiates and the role of Nanog in this process. To do this, they make use of previously generated mESC lines, including a Dox-inducible mutant Nanog line (N51A) that does not bind the regulatory region of PrE genes. Global gene expression was examined after Nanog deletion. As expected, PrE genes became upregulated. TBX3 was also increased in Nanog null ESCs, but could be rescued by provision of exogenous NANOG. ChIP-seq data revealed that most of the NANOG binding was distal to the transcriptional start site. NANOG bound to enhancers with high H3K27ac, H3K4me2 and H3K4me1. Histone H3K36me3 allows access of RNA Polymerase to DNA to which NANOG was found to bind. The authors generated histone modification maps and showed interaction of NANOG with P300 and the Compass complex on enhancers and promoters. Using various experimental procedures, they implicate a role for NANOG in repressing TBX3 via its promoter to control PrE differentiation. As reported previously, the authors show that TBX3^{-/-} cells do not lose pluripotency genes, but Nanog/Tbx3 DKO ESCs exhibited impaired colony formation, providing tentative evidence that NANOG and TBX3 cooperate to repress PrE and maintain ESC self-renewal. Binding of the PRC2 components EED, EZH2 and SUZ12 is reduced in Nanog null cells. As expected, ablation of TBX3 in Nanog null cells reversibly resulted in reduced expression of PrE markers. ChIP-seq was performed for TBX3, but the results were relatively uninformative, particularly since the major finding was associated with limb development. DKO of both Nanog and Gata6 blocked expression of Gata4 and Sox17, but transient OE of Gata6 enabled its binding to their promoter and enhancer regions. Finally, the authors investigated published scRNA-seq of E3.5 and E4.5 mouse embryos to compare the effect of varying levels of Nanog expression on PrE genes. As expected, they found that high Nanog was associated with low Gata6, 4 and Sox17, but high Tbx3. In Nanog low cells Tbx3 was co-expressed with these PrE genes. The authors propose that a poised cellular state is achieved in cells exhibiting medium levels of Nanog expression. As expected, by E4.5 expression levels of epiblast versus PrE genes had resolved to those implicated by immunofluorescence.

Points needing to be addressed:

1. Particularly in the introduction, many of the references regarding derivation of ESCs or lineage segregation in early embryos are reviews. The authors need to cite the original references, which they should be able to find within the reviews. Later on in the text, there is better coverage of experimental history.
2. It would be helpful to include a sentence in the introduction to reference the paper published by Artus, Pilizsek and Hadjantonakis, *Dev. Biol.* 2011, which documents the order of PrE gene activation in the developing mouse blastocyst because it helps to motivate the choice of genes used in the present study for readers less familiar with the field.
3. On P4, the authors state that they made maps for H3K27ac; H3K4me3 and H3K27me3 from their mutant cell line, but in the subsequent analysis there is no further mention of H3K27me3 in the results. Please could the authors explain why?
4. Could the authors please elaborate as to why they are using RCNbh ESCs for Nanog depletion instead of establishing a degron line, which would enable much more rapid reduction of NANOG protein? The authors are obviously familiar with this technology, but chose to use it only for the final part of the study.
5. The reason for the choice of N51A line for some of the experiments needs more explanation.

** As a service to authors, EMBO Press provides authors with the ability to transfer a manuscript that one journal cannot offer to publish to another journal, without the author having to upload the manuscript data again. To transfer your manuscript to another EMBO Press journal using this service, please click on Link Not Available

Dear Martina,

I hope this message finds you well.

First, I would like to sincerely thank you for your time and effort in handling the review of our manuscript. We also deeply appreciate the constructive feedback provided by the reviewers. Their insights have been invaluable to us in refining our work.

After carefully reviewing the comments from the reviewers, we must admit that we are somewhat perplexed by the decision to reject our manuscript. We greatly appreciate the reviewers' thoughtful and constructive feedback, and we are grateful that all three reviewers acknowledged the novelty and potential interest of our research, providing positive comments about the value of our work. Their recognition of the manuscript's merits has led us to believe that our study indeed has significant value. With that in mind, we would like to take this opportunity to further clarify the importance of our work and the contributions it makes to the field.

Both Nanog and Tbx3 are well-established pluripotency factors. Our research provides novel insights by demonstrating, for the first time, that Nanog and Tbx3 form a complex to sustain the self-renewal of mESCs through the repression of endodermal gene expression via newly identified distal enhancers. Furthermore, we discovered that the loss of Nanog induces Gata6 expression in a Tbx3-independent manner, promoting endoderm differentiation. Tbx3 then collaborates with Gata6 to enhance the expression of endodermal genes, driving differentiation initiated by Nanog depletion.

Tbx3 has long been recognized for its dual role in maintaining self-renewal and promoting differentiation in ESCs (Esmailpour and Huang, 2012; Ivanova et al., 2006; Lu et al., 2011; Russell et al., 2015; Weidgang et al., 2013; Zhang et al., 2019; Zhao et al., 2014). However, the molecular mechanism underlying this dual functionality has remained unclear. Our findings provide critical insights, revealing that Tbx3's distinct roles in ESCs and PrE differentiation depend on its interactions with specific partners. In ESCs, Tbx3 and Nanog cooperatively bind promoter and enhancer regions of PrE marker genes, sustaining self-renewal. During differentiation, however, as Nanog levels decline and Gata6 is recruited, Tbx3's functional role shifts from maintaining pluripotency to promoting differentiation.

These results not only elucidate a previously uncharacterized mechanism for Tbx3's bifunctionality but also suggest that this model may extend to other pluripotency-associated factors, such as Esrrb. Emerging evidence highlights Esrrb's dual role in ESC maintenance and PrE differentiation, emphasizing its relevance to broader stem cell biology (Festuccia et al., 2016; Festuccia et al., 2018a; Festuccia et al., 2018b; Wamaitha et al., 2015; Uranishi et al., 2016; Herchcovici Levy et al., 2022). Notably, a recent study corroborates this concept, further underscoring the parallels between Tbx3 and Esrrb in stem cell regulation (Knudsen et al., 2023).

Our findings also propose a robust autoregulatory mechanism that preserves stem cell identity while enabling differentiation in response to developmental cues. Specifically, we observed rapid upregulation of Gata6 following Nanog degradation, which acts synergistically with increased Tbx3 expression to further activate Gata6 and other endodermal genes. This autoregulatory

loop facilitates ESC differentiation toward the endodermal lineage and accelerates PrE differentiation. Exploring whether similar mechanisms apply to other endodermal markers, such as Gata4 and Sox17, could provide deeper insights into stem cell differentiation.

We fully recognize the importance of the reviewers' feedback and assure you that we are committed to addressing each comment thoroughly. In particular, we take Reviewer 2's observation regarding the clarity of the manuscript very seriously. We are confident that we could resolve all the issues raised and therefore enhance the manuscript's quality.

Given the potential impact of our findings and our commitment to addressing all reviewer comments thoughtfully and thoroughly, we would like to kindly request that you reconsider our manuscript for your Journal.

Once again, thank you for your time and consideration. We deeply value your support and look forward to your response.

Best wishes,

Wensheng

Dear Wensheng,

Thank you for your letter asking us to reconsider our decision and invite revision of your manuscript. I apologize for my delayed response, but I have now carefully read your letter and re-read the referee reports. I appreciate that the referees considered your study of potential interest, but I also note that they raised a number of important concerns. In particular referee #2 has significant concerns regarding missing controls and is not convinced that the data - as the manuscript stands - support the proposed model and claims regarding the relative roles of Nanog, Tbx3, and Gata6 in stem cell maintenance and endoderm differentiation. Referee #1 and #2 also point out a number of contradictory results. Referee #3 raises concerns along the same lines and considers the currently provided evidence as "[...] tentative evidence that Nanog and Tbx3 cooperate to repress PrE and maintain ESC self-renewal". The referee raises further concerns that the results of the ChIP-seq data for Tbx3 are mainly associated with limb development and therefore not informative.

As the manuscript stands, neither of the referees is convinced that the data sufficiently support the main conclusions. From the referee comments it is clear that addressing all these concerns requires a significant amount of work with an unclear outcome and I can therefore not reverse our decision and invite revision of your manuscript. That said, we recognize the potential interest of your study, and I am open to reconsider your manuscript, if you were able to address all referee concerns in full and considerably strengthen your work in the near future.

In case you decide to embark on such a revision and re-submit your manuscript, I want to alert you to the fact that we will re-assess novelty taking into account any literature at the time of resubmission.

I am sorry that I cannot be more positive at this stage, but hope that the referee comments will be helpful in strengthening your study.

Kind regards,

Martina

** As a service to authors, EMBO Press provides authors with the ability to transfer a manuscript that one journal cannot offer to publish to another journal, without the author having to upload the manuscript data again. To transfer your manuscript to another EMBO Press journal using this service, please click on Link Not Available

Referee #1:

In this manuscript, the authors reported a novel regulatory cue to direct differentiation of mouse embryonic stem cells (ESCs) into the extraembryonic endoderm (ExEn) cells. Using the inducible knockout ESCs, they revealed that the homeobox transcription factor Nanog represses the expression of the ExEn-specific transcription factors Gata6, Gata4 and Sox17. Nanog also represses the expression of pluripotency-associated transcription factor Tbx3. Interestingly, Tbx3 cooperate with Nanog to repress the ExEn-specific transcription factors. On the other hand, Tbx3 involves in the transcriptional activation of these genes in the absence of Nanog. These results suggested the dual role of Tbx3 in the differentiation to ExEn.

It was reported that the competitive function of Nanog and Gata6 involves in the lineage segregation of epiblast and ExEn. The finding shown in this report demonstrated the regulatory mechanism in more comprehensive manner by accompanying Tbx3. It is potentially worth to be published in EMBO Reports after revision for the points listed in below.

1. The authors use RCN β H ESCs for inducible knockout of Nanog in ESCs to show the induction of ExEn markers after elimination of Nanog. How about the morphological differentiation after elimination of Nanog? Does the the induction of ExEn markers depend on the culture condition i.e external signals? Wah happens in serum-free 2iLIF culture?

We greatly appreciate the valuable comments of the reviewer. To address the question regarding morphological differentiation following Nanog elimination, we conducted experiments to observe morphological changes in RCN β H ESCs under two distinct culture conditions—FBS/LIF and 2i/LIF—after 96 hours of Nanog knockout.

Under the FBS/LIF condition, Nanog knockout resulted in notable morphological alterations. The colonies became smaller and exhibited characteristics indicative of differentiation, in contrast to the undifferentiated morphology of control RCN β H ESCs (Figure A below). This observation aligns with our qPCR analysis, which revealed a substantial upregulation of ExEn markers, with expression levels increasing several hundred-fold upon Nanog deletion, indicating robust differentiation towards the ExEn lineage.

Similarly, under 2i/LIF conditions, significant morphological changes were observed after Nanog knockout. The Nanog-deficient colonies became smaller and exhibited signs of differentiation (Figure B). Consistently, qPCR analysis showed increased expression of ExEn markers, approximately double (Figure C). Several reports shown the inhibition of FGF/MAPK signaling inhibits the ExEn differentiation of mouse ESCs (Nichols et al., 2009; Kang et al., 2013; Schröter et al., 2015). Therefore, the modest increase in ExEn marker gene expression upon the deletion of Nanog in 2i/LIF condition might be resulted from combination of the direct repression of Nanog on ExEn gene expression and the inhibition of FGF/MAPK signaling.

In summary, these findings indicate that Nanog depletion induces ExEn differentiation under both FBS/LIF and serum-free 2i/LIF conditions, but it mitigates the differentiation response specifically under serum-free 2i/LIF conditions.

Figure for referee with unpublished data and its description has been removed upon request by the authors.

2. The phenotype of Tbx3 KO and Nanog: Tbx3 double KO ESCs is important part of this manuscript. Do they undergo morphological differentiation to ExEn? How about the self-renewal capacity in 2iLIF?

We sincerely appreciate the reviewers' insightful comments. To address the issue of morphological differentiation following Tbx3 knockout (KO) and Nanog/Tbx3 double knockout (dKO), we examined the morphological changes in RCN β H and RCN β H-Tbx3 KO ESCs cultured under 2i/LIF conditions at 1, 2, 3, and 4 days post-induction of Nanog knockout. Under 2i/LIF conditions, Nanog deletion induced marked morphological changes; compared with the undifferentiated morphology of control ESCs (-4-OHT RCN β H), Nanog-deficient colonies became smaller and displayed signs of differentiation (+4-OHT RCN β H, Figure A). Similarly, Tbx3 knockout alone resulted in notable morphological alterations, with RCN β H-Tbx3 KO colonies exhibiting reduced size and differentiation features (-4-OHT RCN β H-Tbx3 KO, Figure A). Notably, cells subjected to Nanog/Tbx3 double knockout exhibited an enhanced tendency toward differentiation. These observations are consistent with our qPCR results (Figure B), which showed that individual deletion of Nanog or Tbx3 impaired ESC self-renewal. Double knockout of Nanog and Tbx3 led to a more pronounced downregulation of core pluripotency genes, including Oct4, Sox2, Klf4, Klf2, and Klf5, indicating a more severe

Figure for referee with unpublished data and its description has been removed upon request by the authors.

3. The role of PRC2 on ExEn differentiation has not been described in the references cited by the authors. It should be assessed in their ESC models with Ezh inhibitor.

We thank the reviewer for the insightful comments regarding the role of PRC2 in ExEn differentiation. In the cited references, the repressive role of the PRC2 component EED on the expression of Gata6, Gata4, Sox7, and Sox17 has been demonstrated. Specifically, deletion of Eed led to significant upregulation of these ExEn genes, as reported in Boyer et al. (Nature, 2006; Figure 2) and in our previous work (Zhang et al., Cell Stem Cell, 2019; Figure 5).

To further address the reviewer's suggestion, we examined the role of PRC2 in our ESC model using GSK126, an EZH2 inhibitor. Treatment with GSK126 for 4 hours resulted in upregulated expression of Gata6, Gata4, and Sox17 (Figure below). Consistent to the result from the current manuscript (Figure 4A), NANOG degradation via IAA treatment for 4 hours selectively increased Gata6 expression, with no significant effect on Gata4 or Sox17. Notably, combined treatment with both IAA and GSK126 led to a synergistic increase in Gata6 expression compared to either treatment alone (Figure below).

These findings suggest that Nanog and PRC2 independently and collaboratively regulate ExEn gene expression.

qPCR analysis of relative mRNA expression levels of Gata6, Gata4, and Sox17 in Nanog-mAID ESCs following 4-hour treatments. Experimental groups included untreated control (WT), indole-3-acetic acid treatment (+IAA), GSK126 treatment (+GSK), and combined treatment with IAA and GSK126 (+IAA+GSK).

4. Waghray et al (PMID:26095607) reported that the Tbx3 KO embryos possesses slightly higher numbers of ExEn cells at E4.5, suggesting that its function is dispensable for differentiation of ExEn. How is it consistent to the idea that Tbx3 facilitates ExEn differentiation shown by the authors?

We thank the reviewer for highlighting the important study by Waghray et al. (PMID: 26095607). In their work, Figure S3F from Waghray et al. (2015) shows that Tbx3 KO embryos exhibit a slight reduction in the number of ExEn cells (GATA6+) (Figure below), suggesting that Tbx3 plays a

REDACTED: Figure S3F from Waghray et al, 2015 (PMID 26095607)

supportive, rather than an essential, role

Figure S3F from Waghray et al., 2015 (PMID:26095607)

in ExEn differentiation at E4.5. This observation is consistent with our result that Tbx3 facilitates ExEn differentiation.

Our in vivo data analysis further support this conclusion. Single-cell RNA sequencing analysis (Figure 4P) and TBX3 immunostaining (Figure S4N) in E4.5 embryos reveal high expression of Tbx3 in ExEn cells, reinforcing the notion that Tbx3 positively regulates ExEn differentiation.

Although Waghray et al. (2015) report a slight decrease in ExEn cell numbers at E4.5 (Figure S3F in Waghray et al.' paper), we interpret this as evidence that Tbx3 promotes, but is not absolutely essential for, ExEn differentiation. In our experiments, we observed in Figure 3D that the expression of Gata6, Gata4, and Sox17 in Nanog:Tbx3 double KO cells was reduced compared to Nanog KO cells. However, in Figure 4E, the expression of Gata4 and Sox17 in Nanog:Gata6 double KO cells was nearly completely abolished relative to Nanog KO cells. These findings underscore our conclusion that Tbx3 plays a positive, but not critical, role in ExEn differentiation, in contrast to the essential role of Gata6.

We sincerely appreciate the reviewer's thoughtful comments, which have significantly contributed to the improvement of our manuscript.

5. In Fig 3H, the authors showed the result of enriched motifs at the Tbx3 binding sites. Why is the T-box motif missing? Doesn't it bind to specific DNA sequence directly? The ChIP-seq of Tbx3 in limb bud showed the enrichment showed the enrichment of Eomes (T-box) motif, suggesting the direct binding of Tbx3 to the specific DNA sequence.

We sincerely thank the reviewer for highlighting the recent ChIP-seq study of Tbx3 in the limb bud (Soussi et al., 2024) and for pointing out the question regarding the absence of the T-box motif in our analysis of enriched motifs at Tbx3 binding sites in Figure 3H.

In our analysis, motifs reported by Soussi et al., such as Zfp281, En1, EOMES (T-box), Brachyury (T-box), REST, RUNX2, and ZNFs, were indeed identified. Our *de novo* Motif Enrichment results also confirmed the enrichment of Tbx6, REST, RUNX2. These results align with the findings reported by Soussi et al.

To provide a more comprehensive representation, we have revised Figure 3H and updated the corresponding text in the manuscript. We have also cited the study by Soussi et al. (2024) to acknowledge their findings.

We greatly appreciate the reviewer's insightful comment, which has helped to clarify and strengthen our analysis.

De Novo Motif Enrichment	Tbx3 sites \log_{10} P-value	Best Match TF
	-2.064e+03	Pou5f1::Sox2
	-1.671e+03	ZNF148
	-4.144e+02	ESRRB
	-4.140e+02	TBX6
	-2.069e+02	REST
	-1.846e+02	RUNX2

6. Page 6: The authors stated that Nanog represses Gata6 expression by competitively inhibiting P300 binding to the DE region. What does it mean?

We thank the reviewer for their insightful comment. We apologize for not clearly explaining our data and conclusions in the original manuscript.

The activation of the distal enhancer (DE) using dCas9-VPR in combination with specific guide RNAs targeting the Gata6 DE region significantly upregulated Gata6 expression (Figure 3O). Conversely, deletion of the DE using CRISPR/Cas9 resulted in reduced Gata6 expression in Nanog KO cells (Figure 3P). These results support the regulatory role of the DE in controlling Gata6 expression.

In Nanog KO cells, we observed increased P300 binding and a corresponding elevation in H3K27ac deposition at the DE of the Gata6 gene (Figure 3N). This indicates that NANOG binding at the DE of Gata6 in WT ESCs prevents P300 from binding and thereby inhibits H3K27ac deposition. Thus, we concluded that NANOG represses Gata6 expression by competitively inhibiting P300 binding to the DE region in WT ESCs.

7. Page 11, middle: Zfp81 should be Zfp281.

We apologize for the typographical error and thank the reviewer for bringing it to our attention. The incorrect "Zfp81" has been corrected to "Zfp281" in the revised manuscript.

Referee #2:

In this paper, Wu et al. define a gene regulatory network composed of Nanog, Tbx3 and Gata6, whose interplay modulates the balance between embryonic stem cell (ESC) self-renewal and primitive endoderm (PrE) differentiation. As Tbx3 is a pluripotency factor, the paper suggests it is able to both promote stem ESC self-renewal by suppressing expression of PrE markers and drive PrE differentiation, depending on its context. In particular, association of Tbx3 with Nanog and PRC2 stimulates the repression of PrE transcription. In absence of Nanog, Tbx3 is able to form a complex with Gata6, which in turn activates the expression of PrE-promoting genes.

While the a number of previous studies have highlighted a role for TBX3 in both pluripotency and differentiation, the mechanism by which it acts and the relevance of transcription stoichiometries has not been explored. For this reason, this paper would be suitable for EMBO reports following significant revision. However, the manuscript is both confusing and missing fundemanetal controls. At the level of mechanism, *their model implies TBX3 and NANOG bind DNA together to recruit PRC2, but in Figure 3, they claim NANOG blocks TBX3 binding.* Perhaps this could be addressed by titrating the level of these two factors and assessing their impact on differentiation. *Moroever, they use a NANOG degron, but never show how IAA influences NANOG levels, but always show the influence of IAA indirectly via endodermal transcription or H3K27 modification.* Finally, the writing in the manuscript is very confusing and it is unclear how the reconcile the different obseervations contained in the figures or what the final model is meant to be.

A: At the level of mechanism, their model implies TBX3 and NANOG bind DNA together to recruit PRC2, but in Figure 3, they claim NANOG blocks TBX3 binding.

We thank the reviewer for the insightful comments. In our model, we propose that in wild-type ESCs,

TBX3 and NANOG collaborate with PRC2 to inhibit the expression of PrE genes, which is essential for maintaining ESC self-renewal (**Figure 5**, upper panel). Consistent with this model, in the presence of Nanog, deletion of Tbx3 results in reduced expression of PrE genes (**Figure 3A**), while overexpression of Tbx3 prevents the activation of PrE gene expression (**Figure 3B**). Therefore, in the presence of Nanog, Tbx3 functions as a prepotency factor to maintain ESC self-renewal by repressing PrE gene expression (**Figures 3A-3B**).

Upon Nanog deletion, both Tbx3 and Gata6 expression are upregulated (**Figures 1A and 1G; Figure S1H; Figure 1H**), and together, they promote differentiation towards the PrE lineage (**Figure 5**, middle panel).

Upon Nanog deletion, ESC differentiation towards the PrE lineage is initiated. And Tbx3 changes its role from maintaining the self-renewal of ESCs to promote the expression of PrE genes, in collaboration with Gata6. Following Nanog deletion, Tbx3 expression is upregulated, and the TBX3 binding to PrE genes increases (**Figure 3N; Figures S3M and S3N**). Based on these findings, we propose that Nanog serves to restrict TBX3 binding to PrE genes, maintaining TBX3 binding at a controlled level. To avoid the misunderstanding and confusion, we have not claimed that the NANOG blocks TBX3 binding in the manuscript. We apologize for any confusion caused by our initial interpretation and appreciate the reviewer's constructive feedback.

B: Moreover, they use a NANOG degron, but never show how IAA influences NANOG levels, but always show the influence of IAA indirectly via endodermal transcription or H3K27 modification.

We apologize for any misunderstanding and thank the reviewer for their valuable comments.

The reviewer raised an important point regarding the demonstration of how IAA influences NANOG levels in our study. We would like to clarify that the Auxin-Inducible Degron (AID) technology enables rapid and targeted depletion of proteins in the presence of the small molecule auxin, such as IAA (Nishimura et al., 2009). This system requires the expression of the *Oryza sativa* F-box protein TIR1, which interacts with the native SCF E3 ubiquitin ligase complex to target AID-tagged proteins for degradation (Nishimura et al., 2009).

In our previous study, we generated a Nanog degron cell line by fusing a mini-AID tag to the C-terminus of the Nanog gene in ESCs that overexpress the TIR1 protein (Li et al., 2022). Upon addition of IAA, the interaction between the AID-tagged Nanog protein and the TIR1-SCF E3 ubiquitin ligase complex is induced, leading to efficient degradation of NANOG protein within as little as 2 hours of IAA induction (Figure 2K from Li et al., 2022). The addition of IAA on NANOG degradation was previously demonstrated in our previous work (Li et al., 2022). Therefore, I appreciate the reviewer agree that we have not shown the induced NANOG degradation upon the addition of IAA repeatedly.

To investigate the sequential activation of PrE genes following Nanog deletion, and to explore the underlying epigenetic mechanisms driving Gata6 activation upon Nanog loss, we employed the Nanog degron cell line in this study due to its efficient degradation of min-AID tagged NANOG as short as 2 hours of IAA induction.

We hope this explanation clarifies our experimental approach and we appreciate the reviewer's constructive feedback.

C: Finally, the writing in the manuscript is very confusing and it is unclear how to reconcile the different observations contained in the figures or what the final model is meant to be.

We are sorry for the confusion caused. Tbx3 has been reported to play a dual role in both maintaining self-renewal and promoting differentiation in ESCs (Ivanova et al., 2006; Lu et al., 2011; Esmailpour and Huang, 2012; Weidgang et al., 2013; Zhao et al., 2014; Russell et al., 2015; Zhang et al., 2019). However, the mechanisms underlying this dichotomous function of Tbx3 in ESC maintenance and its promotion on PrE differentiation are poorly understood. In this work, we revealed the mechanism of the dual roles of Tbx3 on the maintenance of self-renewal and the differentiation of ESCs. In the presence of Nanog, Tbx3 collaborates with Nanog and PRC2 complex to maintain the ESC identity upon repressing PrE gene expression (Upper panel of Figure 5). Upon the deletion of Nanog, tbx3 expression was upregulated and the binding of TBX3 on PrE genes increased, thereby promoting the PrE gene expression with Gata6 (middle panel of Figure 5). In line with the positive role of Tbx3 on the regulation of PrE gene expression in the absence of Nanog, the deletion of Tbx3 impaired the elevated PrE gene expression upon Nanog deletion (lower panel of Figure 5). We wish the detailed response to the comments below could clarify the confusion to the reviewer. We apologize for the misunderstanding caused.

Specific points

• **S11 it is unclear that the relative expression plotted is Tbx3, so adding plot title would help with clarity as they did in Fig. 1G.**

We thank the reviewer for this helpful suggestion. In response, we have added a plot title to clarify that the relative expression shown in S11 represents Tbx3, consistent with the labeling approach used in Fig. 1G. We believe this addition enhances the clarity of the figure.

• **Figure S2D. In the text, this figure is used to make the point that the Nanog-bound sites that lose H3K27ac are predominantly in enhancers ("[...] 40% of sites with an increase in H3K27ac levels and 65% of sites with decreased H3K27ac were bound by NANOG in WT ESCs (Figure 2C and Figure S2D) and located predominantly in enhancers (Figure S2D), indicating that NANOG contributes to the regulation of H3K27ac at active enhancers in ESCs."), but there is no explanation of how these enhancers were defined. Are these sites marked with H3K27ac and H3K4me3 (normally associated with promoters)? Or are these just sites that had H3K27ac in the WT but lose it in the Nanog KO? Why would then these be enhancers? Some clarification is needed.**

We are deeply appreciative of your highly valuable comments regarding the content associated with Figure S2D in our paper. Your feedback plays a pivotal role in enhancing the overall quality of our work.

Concerning the definition of enhancer you brought up, as you accurately noted, H3K27ac serves as a modification marker for both promoters and enhancers. Our original text contained certain unclear and imprecise descriptions. We would like to clarify that the alterations in H3K27ac under discussion here encompass not only those in enhancers but also changes in promoter activity. At present, we have revised the relevant content to guarantee more precise expression and to eliminate any potential ambiguity.

Enhancer regions are typically defined by H3K27ac and H3K4me1 modifications. Thus, we conducted a further analysis of the 631 sites (displaying an increased H3K27ac modification upon Nanog deletion in Figure 2C) and 2366 sites (exhibiting a reduced H3K27ac modification upon Nanog deletion in Figure S2D) using ChIP - seq for H3K4me1 in WT ESCs as per the study by Zhang et al. (2019). Our aim was to investigate whether there is H3K4me1 modification at these altered sites. By doing so, we could determine how Nanog deletion impacts enhancer activity across the genome. Among the 631 sites with a more than 2-fold increase in H3K27ac deposition, 61% (384) were found to have H3K4me1 modification, suggesting that they are enhancer regions. Among the 2,366 sites with a more than 2-fold reduction in H3K27ac modification, 75% (1762) had H3K4me1 modification, indicating their status as enhancer regions. Evidently, Nanog deletion exerts an influence on the enhancer activity of genes.

In conjunction with the H3K4me3 analysis (Figure 2E and Figure S2G), our findings support that Nanog regulates the expression of its target genes by modulating the activity of both promoters and enhancers. As the additional analysis presented below (see figure below) does not alter our conclusions, we have opted not to include it in the main manuscript. Instead, we have revised the relevant text in the manuscript to address the reviewer's concern. However, we would be more than happy to incorporate this analysis into the manuscript should the reviewer deem it necessary. Once again, we sincerely appreciate your thorough review and insightful comments.

Figure (left). Nanog knockout (KO) leads to 631 genomic sites with increased H3K27ac modification overlapping with NANOG binding sites, among which 384 sites (61%) are marked by H3K4me1; Figure (right). Nanog knockout (KO) results in 2366 genomic sites with decreased H3K27ac modification overlapping with NANOG binding sites, of which 1762 sites (75%) are marked by H3K4me1.

• **Fig.2I. Naming the tracks K27ac WT and KO could be misleading as the genotype actually refers to NANOG. I would consider changing the colors/transparency level to distinguish WT/KO or clarifying in a legend.**

Thank you for your careful review of our manuscript. We have revised the figure and its legend accordingly to ensure clarity. We appreciate your insightful suggestion.

• **Fig.S2K No explanation of what DE, R1 and R2 stand for is found in text or figure legend.**

We are truly grateful for your meticulous and professional review. Your valuable comments are of great assistance to us in improving our paper.

Regarding the issue you pointed out that the meanings of DE, R1 and R2 were not explained in Figure S2K, we have revised the figure legend of Fig. S2K.

• **Figure S3D The Tbx3 KO ESCs colonies look fine, but the double KO (Nanog/Tbx3) can't make colonies and seems to lose self renewal. If Nanog regulates Tbx3 in their GRN, why is the double mutant more severe?**

We thank the reviewer for the insightful comments. As noted in previous studies (Ivanova et al., 2006, Nature; Lu et al., 2011, JBC), Tbx3 has been identified as a pluripotency factor, and its downregulation impairs the self-renewal of ESCs. In contrast to critical pluripotency factors such as Oct4 and Sox2, whose deletion leads to complete loss of ESC self-renewal, factors like Tbx3 (Ivanova et al., 2006; Lu et al., 2011), Esrrb (Zhang et al., 2008, JBC; Festuccia et al., 2018, EMBO J.), and Klf2/Klf4/Klf5 (Jiang et al., 2008, Nat Cell Biol.) impair, but do not completely abolish ESC self-renewal. As shown in Figure S3D, the Tbx3 KO ESC colonies appear viable, but exhibit slight morphological differences, including reduced colony smoothness, lower staining intensity, and a higher proportion of fully differentiated colonies. However, in the double KO of Nanog and Tbx3, we observe a more severe phenotype, characterized by the absence of colony formation, indicating a greater impairment of self-renewal. This is consistent with our quantitative data in Figure S3C, where the expression of key pluripotency markers, including Oct4, Sox2, and Klf4, is significantly reduced in the double KO compared to either single mutant. These results support the notion that Nanog and Tbx3 function cooperatively to maintain ESC self-renewal.

We agree with the reviewer that further investigation into the molecular mechanisms by which Nanog and Tbx3 regulate the expression of key pluripotency genes would be a valuable direction for future studies.

• **Figure 3B and S3D. The authors conclude that Tbx3 acts as a repressor of PrE differentiation based on the induction of Gata6, Gata4 and Sox17 upon Tbx3 KO. However, the AP staining in Fig.S3D actually shows that the Tbx3 KO cells are still able to generate stem cell colonies. This implies that it isn't required to suppress endoderm differentiation.**

We thank the reviewer for the thoughtful feedback. As noted above, Tbx3 KO ESC colonies remain viable but exhibit subtle morphological differences, such as reduced colony smoothness and less intense staining. The deletion of Tbx3 leads to upregulation of endodermal markers, including Gata6, Gata4, and Sox17 (Figure 3A), suggesting that Tbx3 functions as a repressor of endoderm differentiation in the context of Nanog expression. However, while the upregulation of these genes upon Tbx3 KO is observed (with less than a two-fold increase in Figure 3A), this does not lead to significant differentiation into endodermal lineages, which may explain why Tbx3 KO cells are still capable of forming ESC colonies, as seen in Figure S3D.

In contrast, Nanog deletion results in a more substantial increase in endodermal gene expression, even in the presence of Tbx3 (Figure 1A). These findings suggest that while Tbx3 does contribute to the repression of endoderm differentiation, Nanog plays a more prominent role in this process. Thus, our data support the hypothesis that Nanog and Tbx3 act collaboratively to suppress endoderm differentiation, with Nanog exerting a stronger repressive effect.

• **Figure S3N, the increase in TBX3 binding upstream of the Sox17 and Gata4 loci is modest, is it statistically significant?**

We sincerely appreciate the reviewer's insightful comments. Upon re-evaluating our ChIP-seq data using the DESeq2 package, as described in the Methods section, we found that the increased TBX3

binding upstream of Gata4 was statistically significant. In contrast, the increase in TBX3 binding at the Sox17 locus did not reach statistical significance. To further validate these findings, we performed ChIP-qPCR analysis. The results demonstrated that deletion of Nanog led to a significant increase in TBX3 binding at the DE regions of both Gata4 and Sox17, as shown in the figure below.

TBX3 levels at Gata4 and Sox17 enhancer regions in RCN β H (Ctrl.) and RCN β Ht (Nanog $^{-/-}$) ESCs, determined by ChIP-qPCR.

• Figure 4A, there is no figure showing downregulation of Nanog in mAID-Nanog ESCs upon IAA treatment, a quick Western Blot to mirror the relevant timepoints of this experiment might be useful. Furthermore, the authors suggest "The earlier initiation of Gata6 expression than both Gata4 and Sox17 suggest that Gata6 may facilitate the upregulation of Gata4 and Sox17 upon Nanog deletion (Figure 4A).", but even after 48 hours in absence of Nanog, the increase in Gata6 is never accompanied by significant levels of Gata4 and Sox17, why?

a: a quick Western Blot to mirror the relevant timepoints of this experiment might be useful.

We thank the reviewer for the valuable suggestion. We appreciate the reference to the work of Li et al. (2022), which demonstrated the rapid degradation of NANOG protein by Western blot (Figure 2K from Li et al., 2022). Since this data was previously published by our group, we chose not to include the Western blot results in the current manuscript to avoid redundancy.

REDACTED: Figure 2K from Li et al, 2022 (PMID 36003152)

Figure 2K from Li et al. (2022)

b: but even after 48 hours in absence of Nanog, the increase in Gata6 is never accompanied by significant levels of Gata4 and Sox17, why?

We thank the reviewer for the thoughtful observation. We acknowledge that the increase in Gata6 expression observed after 48 hours in the absence of Nanog does not coincide with significant upregulation of Gata4 and Sox17. While we do not have a definitive explanation for this phenomenon, we hypothesize that a certain threshold of Gata6 expression may be required to effectively activate Gata4 and Sox17. Alternatively, other factors, in addition to Gata6, may be necessary to initiate or promote the expression of Gata4 and Sox17. Although Gata6 has been shown to positively regulate Gata4 and Sox17, this regulatory relationship may be context-dependent, and additional signaling pathways or transcription factors might be involved in their coordinated expression. This is an intriguing avenue for future investigation.

• Figure 4D, how much time in the absence of Nanog produced increased levels of endoermal transcription. It is very difficult to compare this to the degron lines. Were the degron lines a complete loss of function (see above)? How long are the conditional mutants culture in the absence of Nanog.

We thank the reviewer for these insightful questions. In Figure 4D, Nanog deletion in the conditional ESC line (RCN β H ESCs) was induced by treatment with 1 μ M 4-OHT for 4 days, as described in the figure legend.

The degron lines used in this study allow for a rapid and nearly complete degradation of NANOG protein within 2 hours of IAA treatment (Li et al., 2002), as detailed in our response to the previous comment. These lines were utilized to investigate the initiation of PrE gene expression, and we observed a significant upregulation of Gata6 expression within 2 hours of IAA treatment (Figure 4A). After using Nanog conditional knockout cells and treating them with 4-OHT for 4 days, both the PrE gene and the Tbx3 gene showed an increase (Figure 1 and Figure 4D). In addition, after treating the Nanog degron cell line with IAA for 24, 48, 72, and 96 hours respectively, it was found that the upward trends of the PrE genes and Tbx3 gene were consistent with those in the Nanog conditional knockout cells treated with 4OHT (Figure A-B, below). In conclusion, the cell lines constructed by two different methods both confirm that Nanog in the LIF/serum medium inhibit the expression of the Tbx3 and PrE genes

A-B qPCR analysis of transcript levels for the Gata6, Gata4, Sox17 (A), and Tbx3 (B) genes in Nanog-mAID ESCs treated with IAA for 0h, 24h, 48h, 72h, and 96h.

• Fig 3N, shows a very modest GATA6 binding at its own DE. However, its binding elsewhere seems much more dramatic.

We sincerely appreciate your valuable comments regarding the GATA6 binding pattern in Figure 3N. You pointed out that GATA6 exhibits relatively weak binding at its own enhancer (DE region) while binding more prominently at other regions, such as the DE2 region (Figure below). We fully acknowledge this observation.

Our selection of the DE region for investigation was primarily based on the presence of both GATA6 and TBX3 binding sites within this region. One of the central objectives of our study is to explore the cooperative regulatory effects of GATA6 and TBX3 on the expression of downstream PrE target genes. Given this characteristic, the DE region serves as an ideal model for our study. While GATA6 binding is indeed more pronounced in the DE2 region, it lacks TBX3 binding sites and, therefore, does not align with our research focus.

Following your suggestion, we conducted an in-depth analysis of the DE2 region. Our experimental results demonstrate that, despite the absence of TBX3 binding, the DE2 region plays a significant role in regulating GATA6 expression. Notably, Nanog can modulate GATA6 expression via the DE2 region, revealing that Nanog can regulate GATA6 independently of TBX3, thereby highlighting the diversity of GATA6 regulatory mechanisms. Specifically, activation of this region using CRISPR-Cas9 activator resulted in a marked upregulation of GATA6 expression (Figure below A), providing strong evidence for the role of the DE2 region as an enhancer of GATA6 transcription. Additionally, CRISPR-Cas9-mediated deletion of this region restored the expression of GATA6, GATA4, and Sox17 (Figure below B), which were originally upregulated following Nanog depletion. These new findings further enrich

our understanding of GATA6 regulatory mechanisms and will be incorporated into the revised manuscript to enhance the comprehensiveness and depth of our study, if needed.

Once again, we sincerely appreciate your insightful feedback, which has been instrumental in refining our research.

A. qPCR analysis for transcript levels of for Gata6 in ESCs co-transfected with deadCas9-VPR and gRNAs designed to target the DE2 region of Gata6; B. Transcript levels of Gata6, Gata4 and Sox17 in RCNβH (Ctrl) and RCNβHt (Nanog^{-/-}), and two DE2 KO ESC clones treated with and without 4-OHT based on qPCR.

• Fig S4H shows Tbx3 levels in absence of Nanog remain relatively constant, but this contradicts the data in Figure 1. What do they think is the real result?

Thank the reviewer very much for the insightful comment. We truly appreciate your attention to the details in our manuscript.

Regarding the apparent discrepancy between the data in Fig S4H and Figure 1, there are specific reasons for this. In Fig S4H, we utilized the Nanog-AID cell line, and the treatment duration of IAA was relatively short, with a maximum of no more than 48 hours, as indicated in the Figure S4H. In contrast, in Figure 1, we employed the Nanog conditional knockout cells, and the treatment time of 4-OHT was 4 days (96 hours), as indicated in the figure legend. This difference in treatment time accounts for why there was no change in Tbx3 levels within 48 hours in Fig S4H, while significant changes in Tbx3 were observed after Nanog knockout at 96 hours in Figure 1.

To further clarify this issue, we conducted additional experiments. We treated the Nanog-AID cell line with the inducer for 96 hours. As expected, similar to the results shown in Figure 1, the expression of Tbx3 was upregulated at 72 hours and 96 hours (Figure below).

Once again, thank you for your valuable feedback.

A-B qPCR analysis of transcript levels for the Tbx3 genes in WT (A) and Nanog-mAID (B) ESCs treated with IAA for 0h, 24h, 48h, 72h, and 96h.

• Fig 4M shows high expression of Tbx3 in Nanog-high cells, based on reanalysis of in vivo single cell RNAseq. Based on this analysis, the levels of Nanog and Tbx3 appear correlated, but this would appear in contrast to the observation that TBX3 is expressed at high levels in Nanog mutants (Fig. 1).

Thank you very much for your valuable comments, which have prompted us to think more deeply and analyze our experimental results.

In our in vivo single-cell analysis experiment, we indeed observed that in cells with high expression of Nanog, TBX3 also showed high expression. This indicates that there is a positive correlation between the two in naive-state ESCs corresponding to the embryonic stage at E3.5. To further verify this, we manipulated ESCs under the culture conditions of the naive state (LIF+2i culture condition). After knocking out Nanog, we found that the expression of TBX3 decreased (Figure below). This is consistent with the in vivo experimental results, as well as with the conclusion in our article that Nanog and TBX3 jointly maintain stemness under the condition of stemness maintenance. It is also consistent with the results in other reports where the two are regarded as stemness maintenance factors.

However, after knocking out Nanog in ES cells cultured under the condition containing FCS and LIF shown in the Figure 1, the expression of TBX3 increased. Existing research generally holds that ESCs cultured under the condition of FCS and LIF are different from those in the naive state. They are in a heterogeneous state, in which the expressions of both Nanog and TBX3 are heterogeneous, differing from the relatively homogenous expression patterns of the two in the naive state. ESCs in this state have already begun to exhibit some differentiation potential, while some cells still maintain the naive state. Therefore, the observation that Nanog inhibits the expression of TBX3 (knocking out Nanog leads to an increase in TBX3) under this condition is not contradictory to the positive correlation between the two in the naive state.

This phenomenon is also consistent with another important conclusion of our article, that is, TBX3 has different functions in different states of ESCs. In the naive state, TBX3 and Nanog work together to maintain the stemness of cells, and Nanog positively regulates the expression of TBX3. When cells begin to enter the differentiation state, knocking out Nanog will lead to the differentiation of ESCs into the primitive endoderm. At this time, the function of TBX3 changes to promoting differentiation together with GATA6. The staining results of the embryo at E4.5 also support this point. The expression of TBX3 is absent in the epiblast part and mainly concentrated in the primitive endoderm area (Figure S4N).

In conclusion, our experimental results are not contradictory but rather strongly support the important conclusion in the article that "Nanog and TBX3 work synergistically during stemness maintenance, TBX3 and GATA6 work synergistically during differentiation, and the regulation of TBX3 by Nanog varies depending on the cell state." We believe that the above explanation has clearly addressed your concerns.

Thank you again for your invaluable comments!

Figure for referee with unpublished data and its description has been removed upon request by the authors.

• The manuscript makes multiple contradictory statements about the relationship between Nanog and Tbx3, stating the former represses the latter in the first paragraphs (or, in vitro), while claiming the opposite in the last section of results (in vivo). As a result it is very difficult to discern what the authors mean.

We sincerely thank the reviewer for their insightful comments and for highlighting this potential source of confusion.

Please see the response to the former comment.

• **Finally, it is not clear why the authors do not discuss the analogy to the model in Knudsen et al 2023, but simply refer this as appearing during manuscript preparation as this paper has been in the public domain for sometime.**

We thank the reviewer for this comment. In response, we have incorporated a discussion of the recent work by Knudsen et al. (2023) into our manuscript, aligning with the reviewer's suggestion.

Referee #3:

In this manuscript by Wu and colleagues, the authors use mouse embryonic stem cells to investigate the potential mechanisms by which primitive endoderm differentiates and the role of Nanog in this process. To do this, they make use of previously generated mESC lines, including a Dox-inducible mutant Nanog line (N51A) that does not bind the regulatory region of PrE genes. Global gene expression was examined after Nanog deletion. As expected, PrE genes became upregulated. TBX3 was also increased in Nanog null ESCs, but could be rescued by provision of exogenous NANOG. ChIP-seq data revealed that most of the NANOG binding was distal to the transcriptional start site. NANOG bound to enhancers with high H3K27ac, H3K4me2 and H3K4me1. Histone H3K36me3 allows access of RNA Polymerase to DNA to which NANOG was found to bind. The authors generated histone modification maps and showed interaction of NANOG with P300 and the Compass complex on enhancers and promoters. Using various experimental procedures, they implicate a role for NANOG in repressing TBX3 via its promoter to control PrE differentiation. As reported previously, the authors show that TBX3^{-/-} cells do not lose pluripotency genes, but Nanog/Tbx3 DKO ESCs exhibited impaired colony formation, providing tentative evidence that NANOG and TBX3 cooperate to repress PrE and maintain ESC self-renewal. Binding of the PRC2 components EED, EZH2 and SUZ12 is reduced in Nanog null cells. As expected, ablation of TBX3 in Nanog null cells reversibly resulted in reduced expression of PrE markers. **ChIP-seq was performed for TBX3, but the results were relatively uninformative, particularly since the major finding was associated with limb development.** DKO of both Nanog and Gata6 blocked expression of Gata4 and Sox17, but transient OE of Gata6 enabled its binding to their promoter and enhancer regions. Finally, the authors investigated published scRNA-seq of E3.5 and E4.5 mouse embryos to compare the effect of varying levels of Nanog expression on PrE genes. As expected, they found that high Nanog was associated with low Gata6, 4 and Sox17, but high Tbx3. In Nanog low cells Tbx3 was co-expressed with these PrE genes. The authors propose that a poised cellular state is achieved in cells exhibiting medium levels of Nanog expression. As expected, by E4.5 expression levels of epiblast versus PrE genes had resolved to those implicated by immunofluorescence.

A: ChIP-seq was performed for TBX3, but the results were relatively uninformative, particularly since the major finding was associated with limb development.

We thank the reviewer for their insightful comments. In response to the concerns raised, we would like to clarify that although the ChIP-seq data for TBX3 did not provide extensive new insights, the major findings are still valuable in the context of understanding TBX3's functional role. Specifically, as shown in Figure 3G, we identified that TBX3-binding sites are enriched near genes involved in pluripotency,

cell fate commitment, response to LIF, and maintenance of stem cell populations. These findings are consistent with the well-documented roles of Tbx3 in both maintaining ESC self-renewal and regulating ESC differentiation. Furthermore, our GO analysis revealed that TBX3-occupied regions are associated with genes implicated in limb development, which aligns with previously established roles of Tbx3 in limb morphogenesis. These results support the dual roles of Tbx3 as a transcription factor that not only sustains ESC pluripotency but also participates in the regulation of differentiation processes, including those involved in limb development. Additionally, as an important factor for maintaining the self-renewal of ESCs, TBX3 has previously lacked high-quality ChIP-seq data in ESCs. We have provided high-quality ChIP-seq data, which serves as an important resource for subsequent studies on the role of transcription factors in ESCs.

Points needing to be addressed:

1. Particularly in the introduction, many of the references regarding derivation of ESCs or lineage segregation in early embryos are reviews. The authors need to cite the original references, which they should be able to find within the reviews. Later on in the text, there is better coverage of experimental history.

We sincerely appreciate the reviewer’s thoughtful suggestion. In response, we have revised the introduction to replace the review articles with the original primary research studies, as recommended. We believe this modification significantly enhances the scholarly rigor and precision of the manuscript, providing a more direct reference to the foundational work in the field.

2. It would be helpful to include a sentence in the introduction to reference the paper published by Artus, Pilizsek and Hadjantonakis, Dev. Biol. 2011, which documents the order of PrE gene activation in the developing mouse blastocyst because it helps to motivate the choice of genes used in the present study for readers less familiar with the field.

We thank the reviewer for this insightful suggestion. In response, we have incorporated a sentence into the introduction referencing the study by Artus, Pilizsek, and Hadjantonakis (Dev. Biol., 2011). This reference elucidates the sequential activation of PrE-specific genes in the developing mouse blastocyst, thereby providing critical context for our selection of genes analyzed in this study. We believe this addition enhances the manuscript by offering valuable background information for readers who may be less familiar with this aspect of the field.

3. On P4, the authors state that they made maps for H3K27ac; H3K4me3 and H3K27me3 from their mutant cell line, but in the subsequent analysis there is no further mention of H3K27me3 in the results. Please could the authors explain why?

We sincerely appreciate the reviewers’ insightful comments. In response, we analyzed genome-wide changes in H3K27me3 modification following Nanog deletion in ESCs. We identified 354 sites where H3K27me3 levels more than doubled upon Nanog knockout. Conversely, 873 sites exhibited a reduction in H3K27me3 signal by more than 1-fold (**Figure A**). GO analysis of these

differentially modified regions revealed enrichment in terms related to development, cell differentiation, and related biological processes (**Figure B**). These results (**Figure A**) have been incorporated into the revised Figure S3. We are grateful to the reviewers for their thoughtful suggestions, which have helped strengthen and refine our manuscript.

(A) Heatmap showing normalized tag density profiles of H3K27me3 levels in wild-type (WT) and Nanog knockout (KO) ESCs, highlighting regions with increased or decreased H3K27me3 upon Nanog deletion; (B) GO enrichment analysis of biological processes associated with genomic regions exhibiting increased (top) or decreased (bottom) H3K27me3 levels in Nanog KO ESCs.

4. Could the authors please elaborate as to why they are using RCN β H ESCs for Nanog depletion instead of establishing a degron line, which would enable much more rapid reduction of NANOG protein? The authors are obviously familiar with this technology, but chose to use it only for the final part of the study.

We thank the reviewer for the thoughtful comment. We chose to use RCN β H ESCs (Nanog fl/fl ESCs kindly provided by Ian Chambers) for most of our experiments because these cells provide a reliable model for conditional Nanog depletion (Hall et al., 2009; Festuccia et al., 2012; Gagliardi, et al., 2013; Papanayotou et al., 2014; Novo et al., 2016; Li et al., 2022). While we are indeed familiar with the Nanog degron system, which allows for the rapid degradation of NANOG protein (within 2-4 hours of IAA induction), we opted to use the degron system selectively in this study to address specific questions.

The degron line was employed to examine the rapid upregulation of Gata6 expression upon NANOG degradation, which occurs as early as 2 hours post-IAA treatment (Figure 4A). However, Gata4 and Sox17 expression were not significantly upregulated even after almost 2 days of NANOG degradation, supporting our conclusion that Gata4 and Sox17 are regulated downstream of Gata6.

Additionally, the degron line was used to investigate the dynamics of histone modifications (H3K27ac and H3K27me3) following Nanog depletion (Figures 4B-4C), as this system allows for precise and rapid changes in NANOG protein levels.

In response to the reviewer's concern, we have also examined the expression of PrE genes and Tbx3, upon IAA induction at multiple time points (24, 48, 72, and 96 hours) to provide a more comprehensive view of the temporal dynamics of Nanog depletion in the degron line. In consistent to the results from RCN β H ESCs (Figure 1 from the manuscript), the degradation of NANOG upon IAA induction led to the upregulation of PrE genes Gata6, Gata4, and Sox17, and Tbx3 gene (Figure below).

We hope this clarification addresses the reviewer's inquiry.

A-B qPCR analysis of transcript levels for the *Gata6*, *Gata4*, *Sox17* (A), and *Tbx3* (B) genes in WT ESCs treated with IAA for 0h, 24h, 48h, 72h, and 96h; C-D qPCR analysis of transcript levels for the *Gata6*, *Gata4*, *Sox17* (C), and *Tbx3* (D) genes in Nanog-mAID ESCs treated with IAA for 0h, 24h, 48h, 72h, and 96h.

5. The reason for the choice of N51A line for some of the experiments needs more explanation.

We thank the reviewer for raising this important point regarding the rationale for selecting the N51A line for certain experiments. The N51A line expresses a mutant version of NANOG with a compromised DNA-binding domain, making it a valuable tool for studying the functional importance of NANOG's DNA-binding activity. Specifically, we used this line to investigate how the loss of DNA binding affects the regulation of specific target genes. In response to the reviewer's suggestion, we have added further clarification in the manuscript to explain this choice in more detail.

Dear Wensheng,

Thank you for the submission of your revised manuscript to EMBO reports. I sincerely apologize for the delay in handling your manuscript, but we have meanwhile received the comments from the referees that were asked to assess it.

I have carefully assessed the comments and am sorry to say that the evaluation of your manuscript is not a positive one. As you will see, all three referees acknowledge that you have strengthened your manuscript to some extent but neither of the referees is fully convinced by the current data set.

Given the number of remaining concerns, the fact that you already had a chance to significantly revise the study and that EMBO reports allows a single round of revision only, I am afraid that we cannot offer to publish the manuscript at this point.

I am sorry that I could not bring better news this time and hope that the referee comments will be helpful in your continued work in this area.

Kind regards,

Martina

=====

Referee #1:

In this revised manuscript, the authors made significant revision for most points that the reviewers mentioned. Overall, the quality of the manuscript now satisfies the level for the publication.

On the other hand, the matter of the morphological differentiation event is not fully addressed. The authors showed the morphological change of ES cells after extinction of Nanog for the reviewers. Since this is important to indicate the real differentiation event, it should be shown in the main figure. I think that the up-regulation of the marker genes is essential but not sufficient to prove the proper differentiation event. However, for this purpose, it is still unclear whether they mimic the differentiation to extra-embryonic endoderm as in the case of the over-expression of Gata6 shown in Fig S3R. Even if the Nanog-null ES cells do not fully mimic the differentiation induced by Gata6, it should be clearly shown followed by the discussion why such difference take place. Since the over-expression of Gata6 induces differentiation to XEN cells (Fujikura et al, Genes Dev 2002), up-regulation of Gata6 in Nanog-null ES cells could sustain proper differentiation event. Therefore, if it is not the case, possible cause should be argued. From the beginning of the research of Nanog function, there is controversy for the phenotype of Nanog-null ES cells. Mitsui et al showed that their Nanog-null ES cells undergo differentiation to extra-embryonic endoderm-like cells with up-regulation of endoderm marker genes including Gata4 and Gata6 (Cell, 2003). In contrast, Chambers et al showed that Nanog-null ES cells can be maintained with keeping pluripotency using the inducible KO ES cell line RCNbH that the authors used in this manuscript (Nature, 2007). The discrepancy could due to the different genetic background of the ES cell lines they use, the different culture condition, or the presence or absence of MEF. Here the authors showed that Nanog-null ES cells undergo differentiation to extraembryonic endoderm using RCNbH, it is important to show how much degree they show differentiation phenotype based on their heterogeneity, morphology and marker gene expression level in comparison to bona fide ES-derived XEN cells.

Referee #2:

While the authors have tried to address our comments, the manuscript itself appears to have changed little in response to them (unless I am mistaken). If the authors could submit a revised manuscript that shows how they have addressed our points, then I could better judge how the authors have clarified the issues raised about the paper. There appears a degree of a misunderstanding of the review process, that I believe reflects lack of experience and therefore think the authors should be given a chance to address our points within the manuscript, not just in their response document. There are a number of issues about clarity of claims that we raised in the first round of review that are probably best illustrated in the comments below.

1. Relative levels of Nanog and Tbx3 in degnon and rTA responsive mutants:

Every set of lines and experimental conditions are different. They need to show loss of Nanog function in the context tested here. They should provide a Westerns in this manuscript demonstrating Nanog and Tbx3 protein levels in the different contexts (in a the time courses used), in the respective cell lines and culture regimes, a figure from a Frontiers manuscript published three years ago for the degnon is not sufficient.

2. The writing in the paper is confusing: They have responded into our critiques, but not indicated how they have rewritten the text to make the manuscript more succinctly express these ideas. For example, they state that Nanog deletion results in Tbx3 independent GATA6 expression and later in the text Nanog deletion results in the upregulation of Tbx3 that interacts with Gata6 and thereby stimulate its expression. Which is it? Are they distinct temporal waves of regulation?

3. In general, the manuscript needs refining and some more systematic discussion/analysis and this would help them to address the questions we raised in our first review. Why the focus on Tbx3 as opposed to anything else, was it unique in the pluripotency network in responding to Nanog? They claim it emerged as the most significantly impacted pluripotency factors, but they only show Oct4 and Sox2 in the figures.

4. There needs to be some discussion of the relative differences in Tbx3 binding. Why should Nanog bind cooperatively with Tbx3 and yet restrict its binding. Perhaps TBX3 binding is stimulated by cooperativity with Gata6 more so than by Nanog and that Nanog inhibits Gata6 binding. The response to the reviewers comments' suggested the authors removed the statements about Nanog inhibiting Tbx3 binding, but they still appear in the text (at least as far as I can see).

5. Our point about Gata6 expression and no Sox17 etc. The authors really need to think about this and explain it in their text. In the response figure, not included in the paper, they show robust Gata6 expression in the longer degnon time points, and now they have induction of Sox17. Why is this not included to clarify the point?

6. In response to our point about Nanog vs Tbx3 correlation. In the current manuscript the discussion of the Nang low population in vivo requires some attention. They say that Nanog low at E3.5 don't co-express endoderm and Tbx consistent with a priming role, but then at E4.5 they do, inconsistent with a priming role. In their response to our comments' they explore the effect of Nanog on Tbx3 in 2i/Lif and compare it to serum, claiming based on RT-PCR that the differences in expression are due to heterogeneity. To address distinct modes of Nanog regulation on Tbx3 in different populations, they really need some form of single cell assay, IF or flow cytometry.

7. The response and the current manuscript both contain reference to recruitment of the Cas9-VP16 demonstrating the likely identity of an enhancer. However, the recruitment of this factor to any accessible region near a gene will probably activate transcription and the authors need to consider their interpretations more carefully.

In summary, the authors clearly have a very interesting observation, they just need to be more systematic about how they present it.

Referee #3:

The manuscript has been improved somewhat, although not all reviewers' comments were considered. There are quite a few typos that will need to be corrected. However, the work represents a potential advance in understanding of how NANOG and TBX3 may regulate primitive endoderm specification and regulation in the mouse embryo. They have made use of much of the available sequencing data to help interpret their findings.

** As a service to authors, EMBO Press provides authors with the ability to transfer a manuscript that one journal cannot offer to publish to another journal, without the author having to upload the manuscript data again. To transfer your manuscript to another EMBO Press journal using this service, please click on Link Not Available

Dear Martina,

We sincerely appreciate the time and effort you and the editorial team have devoted to our manuscript, and we are grateful for the constructive feedback from all three reviewers. Their insightful critiques and valuable suggestions will undoubtedly help strengthen our work.

We were encouraged by their recognition of the manuscript's improvements and potential: Reviewer 1 noted that "the authors made significant revisions for most points" and that "the quality of the manuscript now satisfies the level for publication"; Reviewer 2 recommended that "the authors should be given a chance to address the issues"; and Reviewer 3 acknowledged that "the manuscript has been improved somewhat."

After carefully analyzing the comments, we are confident that all remaining concerns can be addressed comprehensively within the scope of a minor revision, which we can complete within 1–3 months. Our plan is as follows:

Reviewers 1 and 3: We have addressed their comments in detail in our point-by-point responses (file attached). We believe our clarifications resolve their remaining concerns, and we do not anticipate further issues.

Reviewer 2:

We have clarified points that may have arisen from misunderstandings of the degon system, providing additional experimental context.

We will implement the suggested adjustments to data presentation (e.g., relocating certain results to the main text) to improve clarity.

We will promptly perform the proposed Western blot analyses and include the results. For the suggested single-cell assays (e.g., immunofluorescence), we are willing to conduct these experiments within the revision period, while noting that they may not fully address the broader mechanistic questions, which we plan to explore in future studies.

Notably, the reviewers' comments primarily focus on contextual refinements rather than on the validity or rigor of our findings. We remain deeply appreciative of the opportunity you extended for revision, especially given that approximately ten months have passed since our initial submission.

We respectfully request that our revised responses be shared with the reviewers for consideration. Should they find our clarifications and planned revisions satisfactory, we are ready to implement all necessary changes promptly to finalize the manuscript. We are also open to any further suggestions from the editorial office, including the possibility of involving an additional reviewer if deemed appropriate.

We remain committed to ensuring the revised manuscript meets the high standards of EMBO Reports and sincerely hope to be given the opportunity to complete these final improvements. Thank you once again for your time, consideration, and support.

Yours sincerely,

Wensheng

Dear Wensheng

Thank you for your email asking us to reconsider our decision and invite a second round of revision.
Thank you also for the further clarification on the revisions you have already implemented from September 17th.

To recapitulate our consultation, I would like to repeat that I had re-read the referee reports and your point-by-point response. As I noted, neither of the referees was fully convinced that the previous revision has sufficiently strengthened the data and your conclusions. I however also note that referee 2 was supportive of a second round of revision, noting that it was often not clear what has been addressed only in the point-by-point response and what had been incorporated in the manuscript. I agree with this assessment and had a similar experience when reading the response.

As I told you, I have overall no objection to consider a further revised manuscript that will be seen again by the referees. I herewith made a decision on your manuscript, that will allow you to upload the revised manuscript files and a detailed point-by-point response. Please make sure to submit a marked-up manuscript, either with track-changes or color-coded text. This will greatly ease and accelerate the process and also make it clear for the referees, where the proposed changes were implemented.

I look forward to seeing a revised form of your manuscript when it is ready.

Kind regards,

Martina

Responses to Reviewer 1's comments:

We sincerely thank the Reviewer for the thoughtful and insightful comments regarding the morphological differentiation events. We apologize for any confusion our original manuscript may have caused.

In this study, we did not aim to induce differentiation of Nanog-deficient ESCs into XEN cells. Our observations of Nanog-deficient ESCs were restricted to short-term 4-OHT treatment (up to 4 days) in a LIF-containing medium, without passaging. Under these conditions, Nanog knockout led to the upregulation of PrE marker genes (Figure 1A), while core pluripotency markers, such as Oct4, remained largely unchanged (Figure S1H), indicating impaired self-renewal rather than full PrE differentiation.

We employed the RCNbH ESC line, kindly provided by Dr. Ian Chambers, and followed the culturing protocols established by his group. Consistent with their findings, Nanog-null ESCs can be maintained in LIF/serum culture medium. However, whether these Nanog-null ESCs can undergo complete differentiation into XEN cells, comparable to Gata6-overexpressing cells in Figure 4E (original Figure 3Q) was not the focus of our current investigation and remains an open and interesting question for future studies.

In contrast, for directed XEN cell differentiation, we followed an approach analogous to the classic work by Fujikura et al. (2002, *Genes & Development*), who overexpressed Gata6 in WT ESCs. Specifically, we used WT ESCs and forced Gata6 or Gata4 expression via its enhancer activation, rather than transgene overexpression. These cells were cultured for 7 days with 3–4 passages in LIF-free medium. As described in the manuscript: “Upon activation of the Gata6 (upper panel, Figure S4E; original Figure S3Q) or Gata4 (lower panel, Figure S4E) enhancers, ESCs displayed morphological characteristics of extraembryonic endoderm (ExE) cells after culturing and passaging in ESC medium without LIF (Figure S4E). Immunostaining for GATA6 and GATA4 confirmed the generation of ExE cells (Figure 4E and Figure S4F; original Figure 3Q and S3R).” This enhancer activation strategy yielded relatively pure XEN cells, as validated by both morphological criteria, IF and typical marker expression.

We have noticed the discrepancy regarding the phenotype of Nanog-null ESCs from the separate studies (Chambers et al., 2003; Mitsui et al., 2003). The corresponding author of current study previously worked with Dr. Ian Chambers, and our observations using the same RCNbH ESC line and culture conditions are in agreement with Chambers et al. (2007, *Nature*), who demonstrated that Nanog-null ESCs can self-renew under specific conditions. This contrasts with the findings of Mitsui et al. (2003, *Cell*), who reported spontaneous differentiation of Nanog-null ESCs into extraembryonic endoderm-like cells. As the Reviewer rightly noted, such discrepancies may stem from differences in ESC lines—E14 in the Chambers study and MG1.19 in the Mitsui study—as well as subtle variations in culture conditions.

We greatly appreciate the Reviewer's insightful and constructive comments, which helped us clarify and contextualize our findings. In response, we have incorporated the morphological changes observed in Nanog-deficient ESCs into the main figures and provide detailed descriptions of the culture conditions in the figure legends to better highlight this key observation. Below are the detailed revisions implemented in response to your suggestions:

1. Incorporation of morphological changes in Nanog-deficient ESCs. We have added the morphological changes observed in Nanog-deficient ESCs to the main figures (Figure 1B). Corresponding descriptions of these morphological features have been supplemented in the main

text (Page 2, highlighted in yellow).

2. Supplementation of Ezh2 inhibitor-related content. In response to the comment on the function of the Ezh2 inhibitor raised in the first round of review, we have incorporated relevant results into Figure 3P (Page 5, highlighted in yellow).

3. Revision of motif analysis description in Tbx3 binding sites. Regarding the question about the absence of the T-box motif in the analysis of enriched motifs at Tbx3 binding sites (Figure 3F), we have revised the description in the main text to address this concern (Page 4, highlighted yellow).

4. The gene name "Zfp81" has been corrected to "Zfp281" throughout the manuscript, with the revised instance in the main text highlighted in yellow (Page 4).

5. We have conducted a full-text revision to improve language clarity, logical flow, and consistency.

Responses to Reviewer 2's comments:

1. Relative levels of Nanog and Tbx3 in degron and rTA responsive mutants:

Every set of lines and experimental conditions are different. They need to show loss of Nanog function in the context tested here. They should provide a Westerns in this manuscript demonstrating Nanog and Tbx3 protein levels in the different contexts (in a the time courses used), in the respective cell lines and culture regimes, a figure from a Frontiers manuscript published three years ago for the degron is not sufficient.

We sincerely thank the Reviewer for the thoughtful suggestion to directly demonstrate the protein levels of Nanog and Tbx3 under various experimental conditions.

As shown in Figure 1G (original Figure 1F), we have already provided the relative protein levels of Nanog and Tbx3 in RCN β H ESCs cultured under FBS/LIF conditions. In response to the Reviewer's recommendation, we have extended our analysis by including additional Western blot data for NANOG and TBX3 in RCN β H ESCs treated with 4-OHT for 1d, 2d, 3d, and 4d (Figure S4M).

The construction and validation of the mAID-Nanog ESC line, including detailed time-course Western blots demonstrating the efficiency of Nanog degradation, were previously described in our 2022 publication (Li et al., 2022). Furthermore, another study from our group utilizing the same mAID-Nanog ESCs is currently under its second round of review at Nature Communications, where the reproducibility and reliability of this system have been positively recognized. Nevertheless, we fully agree that including these data in the present manuscript will enhance its clarity and rigor. Therefore, we have incorporated Western blot results directly into the main figures, showing both the degradation kinetics of Nanog and the corresponding dynamics of Tbx3 protein levels under the specific experimental conditions used in this study (Figure S3N/S3R).

We are grateful to the Reviewer for this constructive suggestion, which has undoubtedly improved the completeness and transparency of our work.

2. The writing in the paper is confusing: They have responded into our critiques, but not indicated how they have rewritten the text to make the manuscript more succinctly express these ideas. For example, they state that Nanog deletion results in Tbx3 independent GATA6 expression and later in the text Nanog deletion results in the upregulation of Tbx3 that interacts with Gata6 and thereby stimulate its expression. Which is it? Are they distinct temporal waves of regulation?

We are sorry for the potential ambiguity in our textual expression, which may stem from the

involvement of distinct temporal phases in the regulatory process. Our study aims to dissect the dynamics of downstream gene expression following Nanog depletion, thereby clarifying the molecular mechanisms by which Nanog restrains differentiation.

To address this, we employed the degron system and found that Gata6 expression begins to increase as early as 2 hours after Nanog depletion (Figure 3N; original Figure 4A), whereas Tbx3 expression remains largely unchanged during this early phase (Figure S3N; original Figure 4H). These data support a model in which Nanog depletion induces Gata6 expression independently of Tbx3. Mechanistically, this phase is accompanied by reduced H3K27me3 modification at the Gata6 locus, relieving transcriptional repression (Figure 3O; original Figure 4C). Consistently, Nanog depletion in Tbx3-knockout cells still leads to increased Gata6 expression (Figure 4A; original 3D), albeit to a lesser extent than in cells expressing Tbx3. Together, these results indicate that the initial induction of GATA6 expression following Nanog depletion does not require Tbx3.

At a later stage, Nanog depletion results in gradual upregulation of Tbx3 (Figure S4L-S4M; original Figure 3D). TBX3 physically interacts with GATA6 (Figure 4J; original Figure 4H) and co-binds to the Gata6 enhancer (Figure 4B; original Figure 3N). Notably, Nanog knockout enhances TBX3 occupancy at this enhancer (Figure 4B; original Figure 3N), establishing a positive feedback loop that further amplifies GATA6 expression. Supporting this, in Tbx3-knockout cells, reintroduction of TBX3 followed by Nanog depletion yields higher Gata6 expression than in knockout cells without TBX3 rescue (Figure 4A; original Figure 3D), confirming that Tbx3 synergizes with GATA6 to promote its expression during the late phase.

3. In general, the manuscript needs refining and some more systematic discussion/analysis and this would help them to address the questions we raised in our first review. Why the focus on Tbx3 as opposed to anything else, was it unique in the pluripotency network in responding to Nanog? They claim it emerged as the most significantly impacted pluripotency factors, but they only show Oct4 and Sox2 in the figures.

Previous studies have established that Tbx3 possesses dual functional properties: it is involved not only in maintaining the self-renewal of ESCs but also in promoting ESC differentiation under certain conditions. Specifically, Tbx3 has been identified as a key pluripotency factor that supports self-renewal (Ivanova et al., Nature 2006), while also being implicated in facilitating lineage specification (Weidgang et al., Stem Cell Reports 2013; Zhang et al., Cell Stem Cell 2019). This functional duality presents an intriguing mechanistic question: how can a single transcription factor mediate such contrasting outcomes? What regulatory principles underlie this functional switch?

In our study, we observed that the expression dynamics of Tbx3 in Nanog-knockout cells diverged significantly from those of other canonical pluripotency genes. While core factors such as Oct4, Sox2, Klf2, Klf4, and Klf5 (partial data not shown) were either downregulated or remained unchanged, Tbx3 was uniquely upregulated (Figure S1H). Notably, this upregulation coincided with increased expression of PrE-associated genes such as Gata6 (Figure 1A), and is consistent with the previously reported differentiation-promoting function of Tbx3 (Zhang et al., Cell Stem Cell 2019). These findings suggest that Tbx3 may play a distinct and context-dependent role in regulating cell fate following the loss of Nanog.

Based on these observations, we propose a mechanistic model in which the dual functions of Tbx3 are mediated through context-specific interactions with different protein partners. Under the self-renewal conditions, Tbx3 collaborates with Nanog to suppress differentiation-associated gene

expression (Figures 4A-4B; original Figures 3D, 3N). In contrast, in the absence of Nanog, Tbx3 engages alternative partners—such as Gata6—to facilitate PrE differentiation (Figure 4J; original Figure 4H).

4. There needs to be some discussion of the relative differences in Tbx3 binding. Why should Nanog bind cooperatively with Tbx3 and yet restrict its binding. Perhaps TBX3 binding is stimulated by cooperativity with Gata6 more so than by Nanog and that Nanog inhibits Gata6 binding. The response to the reviewers comments' suggested the authors removed the statements about Nanog inhibiting Tbx3 binding, but they still appear in the text (at least as far as I can see).

We sincerely appreciate the reviewer's insightful query regarding the potential relationship between NANOG and TBX3—specifically, whether their interaction is restrictive rather than cooperative, and whether GATA6 may serve as a more dominant binding partner for TBX3.

First, in the context of ESC self-renewal, NANOG and TBX3 primarily act in a cooperative manner to jointly suppress differentiation and sustain pluripotency. Both are core pluripotency regulators, and previous studies have shown that downregulation of either Nanog or Tbx3 led to the increased expression of PrE marker genes (Ivanova et al., Nature 2006; Mitsui et al., Cell 2003; Chambers et al., Cell 2003). Consistent with these reports, we observed that Sox2 and Klf4 expression decreases upon Nanog knockout, and this reduction is further enhanced when Tbx3 is additionally deleted (Figure S3C). Moreover, ChIP-seq analysis indicates that NANOG and TBX3 co-bind to the regulatory regions of major pluripotency factors such as Oct4, Sox2, Klf4, and Klf2. GO analysis of loci showing reduced TBX3 binding after Nanog knockout revealed enrichment for biological processes related to stem cell population maintenance (Figure 3K; original Figure 3M). These observations support a cooperative, rather than mutually restrictive, role for NANOG and TBX3 in maintaining ESC self-renewal.

We also agree with the reviewer's valuable suggestion that “TBX3 binding is stimulated by cooperativity with GATA6 more so than with NANOG” in certain contexts. Upon Nanog knockout, PrE genes such as Gata6 are strongly upregulated (Figures 1A, 3N; original Figures 1A,4A), accompanied by increased Tbx3 expression and markedly enhanced TBX3 binding to the enhancer regions of PrE genes (Figure 4B; original Figure 3N). Importantly, this enhanced binding occurs in the absence of NANOG, consistent with a “partner-switching” mechanism whereby TBX3 transitions from cooperating with NANOG to maintain pluripotency, to partnering with GATA6 to activate PrE genes.

We hope this clarifies the regulatory logic underlying these interactions, and we are grateful to the reviewer for prompting this refined interpretation of the NANOG–TBX3–GATA6 relationship.

5. Our point about Gata6 expression and no Sox17 etc. The authors really need to think about this and explain it in their text. In the response figure, not included in the paper, they show robust Gata6 expression in the longer degran time points, and now they have induction of Sox17. Why is this not included to clarify the point?

We appreciate the reviewer's insightful comment regarding the dynamics of PrE gene expression. In our initial investigations using the rTA-induced Nanog conditional knockout system (RCNβH cells), we observed a clear temporal sequence: upon 4-OHT treatment, Gata6 expression was significantly upregulated as early as day 2, whereas the induction of Gata4 and Sox17 was delayed until days 3–4 (Figure S3P; original Figure S4A). This “Gata6-first activation followed by

Gata4/Sox17 expression” pattern is consistent with the findings of Jérôme Artus et al. (2010, Dev Biol), who demonstrated that Gata4 and Sox17 are expressed subsequent to Gata6.

In experiments using the degron system (Figure 3N; original Figure 4A), we monitored the expression of PrE genes following NANOG degradation at 0, 2, 4, 8, 12, 24, 36, and 48 h. In line with the reviewer’s suggestion, we extended this analysis to 96 h. The results revealed expression trends for PrE genes (Gata6, Gata4, Sox17) and Tbx3 that closely mirrored those observed in RCNβH cells treated with 4-OHT. Specifically, Gata6 responded first, followed by the induction of later genes such as Gata4 and Sox17.

We have incorporated the results from these extended time points into the main text (Figures S3O and S3Q) (Page 5, highlighted in blue) to better illustrate the sequential expression pattern and to avoid the possible misunderstanding that “only Gata6 is expressed while Sox17 is not.”

6. In response to our point about Nanog vs Tbx3 correlation. In the current manuscript the discussion of the Nanog low population in vivo requires some attention. They say that Nanog low at E3.5 don't co-express endoderm and Tbx consistent with a priming role, but then at E4.5 they do, inconsistent with a priming role. In their response to our comments' they explore the effect of Nanog on Tbx3 in 2i/Lif and compare it to serum, claiming based on RT-PCR that the differences in expression are due to heterogeneity. To address distinct modes of Nanog regulation on Tbx3 in different populations, they really need some form of single cell assay, IF or flow cytometry.

We sincerely appreciate the reviewer’s thoughtful and constructive comments.

Regarding the correlation between Nanog and Tbx3: At E3.5, Tbx3 expression is also elevated in Nanog-high cells (Figure 4M), and their co-upregulation contributes to the maintenance of pluripotency. This observation is consistent with our in vitro findings under LIF+2i culture conditions. In contrast, in Nanog-low cells at this stage, Tbx3 expression is low while Gata6 expression is high (Figure 4M), suggesting that the initiation of Gata6 expression occurs independently of Tbx3, which also aligns with our in vitro results.

At E4.5, Nanog-low cells co-express PrE markers together with Tbx3 (Figure 4N; original Figure 4P), consistent with our in vitro observation that Tbx3 and Gata6 can cooperatively promote PrE differentiation. We sincerely appreciate the reviewer’s suggestion to examine Tbx3 expression following Nanog deletion under LIF+2i conditions. While we recognize the heterogeneity of Nanog and Tbx3 expression in ESCs cultured in LIF/serum medium, its precise contribution remains unclear. Moreover, differences in epigenetic landscapes and transcriptional profiles between ESCs maintained in 2i/LIF versus LIF/serum conditions may also account for the distinct regulation of Tbx3 by Nanog. We would be pleased to consider single-cell analyses to further explore this question, although such approaches are resource-intensive. Nevertheless, we anticipate that these experiments may not fully resolve the mechanistic complexity of Nanog-mediated regulation of Tbx3 across distinct cellular contexts. We would greatly appreciate it if the reviewer shares our view that more targeted and refined approaches will be required to delineate these regulatory mechanisms in detail.

7. The response and the current manuscript both contain reference to recruitment of the Cas9-VP16 demonstrating the likely identity of an enhancer. However, the recruitment of this factor to any accessible region near a gene will probably activate transcription and the authors need to consider their interpretations more carefully.

We appreciate the reviewer's constructive comment. The reviewer is correct that recruitment of Cas9–VP16 to any accessible region near a gene may lead to its transcriptional activation, and therefore additional enhancer regions for PrE genes could potentially be identified. In the present study, we focused on identifying and validating candidate enhancer regions for the PrE genes Gata6, Gata4, and Sox17 (Figures 4B, S4A, S4B; original Figures 3N, S3M, S3N). We further confirmed the functional relevance of the Gata6 enhancer region by showing that their deletion, via CRISPR/Cas9, restored the upregulated expression of Gata6 observed upon Nanog deletion (Figure 4D; original Figure 3P). This can further confirm the function of this region.

8. Detailed revisions implemented in response to your comments

We have carefully addressed the specific suggestions raised in your comments and made targeted revisions to the manuscript, with all modified content highlighted in blue for easy identification. The details are as follows:

1. To clearly distinguish the early and late regulatory phases of the studied processes and present our findings more concisely, we have revised the relevant text in the main manuscript (Pages 6–7).

2. We have added a plot title to Figure S11 to explicitly clarify that the relative expression values displayed represent Tbx3.

3. We have revised the legends of the corresponding figures to enhance descriptive clarity, with the updated legends located on Pages 9, 10, 12, and 13.

4. Following your recommendation to contextualize our work with recent literature, we have integrated a discussion of the study by Knudsen et al. (2023) into the main text (Page 8), linking our findings to the latest advances in the field.

5. To optimize the logical flow of the manuscript, we have reorganized the sequence of key results: specifically, we have separated the sections focused on “Nanog and Tbx3 Collaboratively Maintain ESC Self-renewal” and “Nanog and Tbx3 Repress PrE Differentiation” into distinct, thematically coherent parts (Page 4).

6. Finally, we have conducted a thorough full-text revision to refine language expression, improve the precision of scientific descriptions, and ensure consistency in terminology.

Responses to Reviewer 3's comments:

We are very grateful to the reviewer for acknowledging the potential value of this study in understanding the regulation of primitive endoderm specification in mouse embryos by NANOG and TBX3, as well as for affirming our use of sequencing data to interpret the results.

Regarding the reviewer's point that "not all review comments have been considered," we have systematically sorted out all issues in this response and formulated a detailed plan for supplementary experiments and revisions to ensure comprehensive addressing of the concerns. Regarding typos in the text, we have conducted a thorough proofreading during the revision process to strictly standardize the textual expression and improve the rigor and readability of the manuscript.

- 1: Following the recommendation to replace review articles with original primary research studies in the Introduction, we have revised the relevant content. The updated sections of the Introduction are located on Page 2 and highlighted in green, ensuring that our contextualization of the research is grounded in first-hand empirical evidence.

- 2: We have performed an analysis of genome-wide changes in H3K27me3 modification in ESC

following Nanog deletion. The results of this analysis have been integrated into the revised Figure S3K, with corresponding descriptive text added to the main text on Page 5 (highlighted in green) to contextualize the findings.

3: Finally, we have carefully revised the entire manuscript to enhance the logical flow, improve the accuracy of scientific descriptions, and ensure consistency in formatting and terminology. If there are any additional points that we may have inadvertently overlooked, we would be glad to address them promptly.

Dear Prof. Zhang

Thank you for the submission of your revised manuscript to EMBO reports. I apologize for the delay in handling it but we have now received the full set of referee reports that is copied below.

As you will see, both referees find that the study has been significantly strengthened, but referee #2 raises a number of concerns that still have not been resolved adequately. Please address these textual and experimental points and please highlight the changes clearly in a marked up version of the text and in the point-by-point response.

From the editorial side, there are also a few things that we need before we can proceed with the official acceptance of your study.

- 1) Please change the header SUMMARY to Abstract.
- 2) Please provide an email address for the corresponding author on the title page.
- 3) 3-5 Keywords should be provided after the Abstract.
- 4) The Author Contributions need to be removed from the manuscript. Please make sure that the author contributions specified in the online manuscript tracking system are complete and up-to-date. These will be typeset into your article.
- 5) DECLARATION OF INTERESTS should be called Disclosure and Competing Interests Statement.
- 6) The manuscript sections should be in the following order: Title page - Abstract & Keywords - Introduction - Results - Discussion - Methods - Data Availability - Acknowledgments - Disclosure Statement & Competing Interests - References - Figure Legends - (Main Tables with legends if applicable) - Expanded View Figure Legends.
- 7) All main figures need to be provided as separate production quality Figure files; Figure 3 and Figure 4 have two pages which we do not allow. Please either arrange the figures in a manner that all panels fit on one page or split the figure in two.
- 8) You have four supplementary figures. These should be provided in a single PDF called Appendix. The Appendix needs a title page with a table of content and page numbers. The nomenclature is Appendix Figure S1, etc. Each legend should follow its corresponding figure (and the supp. figure legends should be removed from the manuscript). All callouts in the manuscript need to be corrected with the updated nomenclature.
- 9) Table S1 and Table S2 are datasets and should be updated to Dataset EV1 and Dataset EV2 in all places and uploaded as file type Datasets. Please add a legend and the file name as title in a separate tab in the .xls file.
- 10) Please upload the Reagents and Tools table as a separate file and as a Word file instead of .xls. You can download the template from our homepage- Guide to Authors.
- 11) Primary datasets produced in this study need to be deposited in an appropriate public database (see < <https://www.embopress.org/page/journal/14693178/authorguide#dataavailability>>). Specifically, we would kindly ask you to provide public access to the following datasets:
 - ChIP-seq
 - RNA-seq

The accession numbers and database should be listed in a formal "Data Availability " section (placed after Materials & Method) that follows the model below (see also < <https://www.embopress.org/page/journal/14693178/authorguide#dataavailability>>). Please note that the Data Availability Section is restricted to new primary data that are part of this study.

Data availability

- 12) You have reused the data from GSE123046. This dataset should be cited using a data reference. In the text the dataset is cited as (data ref: data reference

Data citations in the article text are distinct from normal bibliographical citations and should directly link to the database records from which the data can be accessed. In the main text, data citations are formatted as follows: "Data ref: Smith et al, 2001" or "Data ref: NCBI Sequence Read Archive PRJNA342805, 2017". In the Reference list, data citations must be labeled with "[DATASET]". A data reference must provide the database name, accession number/identifiers and a resolvable link to the landing page from which the data can be accessed at the end of the reference. Further instructions are available at <<https://www.embopress.org/page/journal/14693178/authorguide#referencesformat>>.

Please note that the paper reporting the dataset can be cited in addition as a normal reference, if applicable.

13) At EMBO Press we ask authors to provide source data for the main manuscript figures. We had already contacted you regarding the source data, but the e-mail might have been overloaded. I will send you the instructions again via a separate email.

Additional information on source data and instruction on how to label the files are available <<https://www.embopress.org/page/journal/14693178/authorguide#sourcedata>>

14) In the Author Checklist line 113 you indicate that information on human clinical and genomic dataset deposition is provided in the manuscript. I think that this does not apply to your manuscript.

15) Once the final version of your manuscript has been submitted, our team of data editors will check the figure legends for accuracy and completeness. To speed up this process, please keep the following in mind:

- The exact p-values must be defined either in the figure legend or in the panels, unless $p < 0.0001$.
- For all quantifications, you need to define the number of replicates (n) and their nature (technical, biological, independent). The bars and error bars must be defined and all quantifications should show the individual datapoints in addition to the mean and error bars.
- Scale bar size should be defined in the legend, not in the figure panel.
- All graphics on images need to be defined, such as arrows, arrowheads, dashed lines etc.

16) The title may not contain punctuation. Could you please provide a draft for a shorter and more concise title?

17) Finally, EMBO Reports papers are accompanied online by

A) a short (1-2 sentences) summary of the findings and their significance,

B) 2-3 bullet points highlighting key results and

C) a schematic summary figure that provides a sketch of the major findings (not a data image).

Please provide the summary figure as a separate file in PNG or JPG format at a size of 550x300-600 pixels (width x height).

Please note that the size is rather small and that text needs to be readable at the final size. Please send us this information along with the revised manuscript.

With kind regards,

=====

Referee #1:

In this revised manuscript, the points raised by this reviewer were well addressed. Now this manuscript is suitable for publication in EMBO Reports.

Referee #2:

The response to our last round of comments, marks an improvement. The authors have addressed a lot of the issues raised. However, it is still not entirely clear how they have modified the text to address the specific points raised in the last round of review. It would really help if they had referred to changes as they respond to comments, rather than at the end of their response document.

Original point 3: They have not really addressed in the paper, only extensively argued in their response document (see above comment). Perhaps they could add a sentence or two to explain why Tbx3, and include expression of more than Oct4, Sox2 and Nanog as representative of all pluripotency genes. When responding to this, provide the editor with line numbers to ensure that it is easy to check the changes.

Original point 4: Please add a few sentences to the discussion clarifying these issues or indicate where they addressed this in the paper.

Original point 6: The embryo images in the paper do not fit with their analysis of clusters. Why should Tbx3 be coexpressed with GATA6 before and after E3.5, but not at E3.5 in the Nanog low cells? The staining in Fig. S4R and S, suggests this is not the case anyway.

Original point 7: As we said previously, the Cas9-VP result doesn't show much, the Cas9 activator only induces expression by 8-old, compared to a huge derepression on Nanog deletion. However, the result needs discussion as the removal of the element is hard to interpret. Is Nanog repressing GATA6 expression or just inhibiting the activity of the enhancer?

A point-by-point response letter to comments raised by the referees of manuscript (EMBOR-2024-60428V5)

Referee #1:

Comment: In this revised manuscript, the points raised by this reviewer were well addressed. Now this manuscript is suitable for publication in EMBO Reports.

Response: We deeply appreciate the reviewer's thoughtful evaluation and the considerable time devoted to our manuscript.

Referee #2:

The response to our last round of comments, marks an improvement. The authors have addressed a lot of the issues raised. However, it is still not entirely clear how they have modified the text to address the specific points raised in the last round of review. It would really help if they had referred to changes as they respond to comments, rather than at the end of their response document.

Comment1: Original point 3: They have not really addressed in the paper, only extensively argued in their response document (see above comment). Perhaps they could add a sentence or two to explain why Tbx3, and include expression of more than Oct4, Sox2 and Nanog as representative of all pluripotency genes. When responding to this, provide the editor with line numbers to ensure that it is easy to check the changes.

Response 1: We sincerely appreciate the reviewer's valuable time and insightful comments on our manuscript. In response to the question regarding our rationale for focusing on the TBX3 gene, we have incorporated additional explanations based on both the research context and our experimental observations, and we have refined the corresponding data accordingly. Our detailed responses are provided below:

Rationale for focusing on TBX3:

We have added a clearer explanation to justify our focus on TBX3. As previously reported, TBX3 plays dual roles in maintaining pluripotency and promoting ESC differentiation. Elucidating the molecular mechanism underlying these seemingly contradictory functions represents a key scientific question addressed in our study and constitutes an important reason for selecting TBX3 as a central target. Furthermore, our experimental results following Nanog knockout revealed that while the expression levels of pluripotency factors Oct4 and Klf5 remained unchanged, those of Sox2, Klf4, and Klf2 were downregulated. In contrast, Tbx3 expression was markedly upregulated at both the RNA and protein levels, accompanied by increased expression of primitive endoderm (PrE) markers. This distinct expression pattern strongly suggests that TBX3 may act as a crucial mediator in Nanog-regulated PrE differentiation, thereby supporting our scientific rationale for focusing on this gene.

Additional experimental data:

In accordance with the reviewer's suggestion, we have supplemented the manuscript with qPCR analyses of additional pluripotency genes—including Sox2, Klf2, Klf4, and Klf5—to provide a more comprehensive overview of the pluripotency gene expression profile after Nanog knockout (now shown in Appendix Fig. S1H).

Revisions to the manuscript text:

We have added relevant explanations on the research background and experimental basis for focusing on TBX3 to the Results section (second paragraph), with clear annotations to facilitate the reviewer's

evaluation.

Once again, we sincerely thank the reviewer for the constructive feedback, which has substantially improved the clarity and completeness of our manuscript. We would be pleased to address any further questions or suggestions.

Comment 2: Original point 4: Please add a few sentences to the discussion clarifying these issues or indicate where they addressed this in the paper.

Response 2: We sincerely appreciate the reviewer's thoughtful advice. In accordance with the suggestion, we have incorporated the relevant explanation (the second paragraph in the Discussion section) into the Discussion section of the manuscript, and the revisions have been clearly marked for ease of review. We are grateful for the reviewer's constructive feedback, which has helped improve the clarity and overall quality of the manuscript.

“Both Nanog and Tbx3 are core pluripotency regulators, and previous studies have shown that reduction of either gene induces ESC differentiation (Ivanova et al., 2006; (Chambers et al., 2003; Mitsui et al., 2003). In agreement with these reports, we observed decreased expression of Klf2 and Klf4 upon Nanog deletion, and this reduction was further exacerbated when Tbx3 was simultaneously deleted (Appendix Fig. S3C). Furthermore, ChIP-seq analysis revealed that NANOG and TBX3 co-occupy regulatory regions of key pluripotency genes, including Oct4, Sox2, Klf4, and Klf2 (Figs. 3H and 3L). GO analysis of loci with reduced TBX3 binding in Nanog-deficient ESCs showed enrichment for biological processes associated with stem cell maintenance (Fig. 3K). Co-immunoprecipitation further confirmed the physical interaction between NANOG and TBX3 (Fig. 4A). Together, these observations support a cooperative role for NANOG and TBX3 in sustaining ESC self-renewal.

In contrast, upon Nanog deletion, PrE genes such as Gata6 become strongly upregulated (Figs. 1A and 4D), coinciding with increased Tbx3 expression and a pronounced enhancement of TBX3 binding at PrE gene enhancers (Fig. 5B). This shift in binding occurs in the absence of NANOG and is consistent with a “partner-switching” model, in which TBX3 transitions from cooperating with NANOG to maintain pluripotency to partnering with GATA6 to activate PrE gene expression. This model provides a mechanistic explanation for TBX3's dual activity and suggests a broader principle whereby pluripotency factors are repurposed to drive lineage specification once core repressors such as NANOG are lost.”

Comment3: Original point 6: The embryo images in the paper do not fit with their analysis of clusters. Why should Tbx3 be coexpressed with GATA6 before and after E3.5, but not at E3.5 in the Nanog low cells? The staining in Fig. S4R and S, suggests this is not the case anyway.

Response 3: We sincerely appreciate the reviewer's insightful and constructive comments. We provide below a concise clarification regarding the consistency between our immunofluorescence (IF) data and the public single-cell RNA sequencing (scRNA-seq) dataset.

Data sources:

The scRNA-seq data were obtained from Nowotschin et al., 2019, whereas the IF experiments were carried out in our study. Minor differences in developmental timing are expected due to biological variation and methodological differences between datasets.

Consistency at 4.5 dpc:

The scRNA-seq data indicate that Tbx3 is co-expressed with primitive endoderm (PE) markers (Gata6, Gata4, Sox17) but not with Oct4 (EPI/ICM). Our IF results are fully consistent with this pattern, showing

TBX3 specifically localized to PE cells.

Clarification at 3.5 dpc:

The 3.5 dpc embryo contains heterogeneous cell populations representing early to late blastocyst states. In the public scRNA-seq dataset, cells were grouped into three subsets based on manually defined empirical thresholds of Nanog expression (rather than absolute quantitative cutoffs):

Nanog-high/Tbx3-high naïve-like cells: lacking PE markers, consistent with pluripotency maintenance.

Nanog-middle cells: moderate TBX3 expression with co-expression of PE markers.

Nanog-low/negative cells: low Tbx3 but high PE markers (Gata6, Sox17, Gata4).

This Nanog-low/Tbx3-low/PE-high subset aligns well with our functional observations: upon Nanog depletion, Gata6 is rapidly induced (Fig. 4D) in a Tbx3-independent manner (Fig. 5A), while Tbx3 upregulation occurs later (Appendix Figs. S4F–S4G). This temporal sequence is consistent with the scRNA-seq profile of the Nanog-low subset.

In our IF analyses, we observed TBX3–GATA6 co-positive cells (may corresponding to the Nanog-middle subset) and TBX3⁺/GATA6⁻ cells (likely the Nanog-high naïve-like subset; Nanog staining was not included here). Although we did not visualize the Nanog-low/Tbx3-low/PE-high population in the current IF experiments, we anticipate that future studies with refined 3.5 dpc staging and targeted marker selection will enable detection of this subset.

Conclusion:

Overall, our IF data, functional assays, and the public scRNA-seq dataset are mutually supportive. Together, they indicate that TBX3 contributes to both PE differentiation (via co-expression with PE markers in Nanog-middle subsets) and pluripotency maintenance (via co-localization with Nanog in naïve cells). The observed complexity at 3.5 dpc reflects the dynamic developmental progression and technical differences between single-cell transcriptomics and IF imaging.

We hope this clarification addresses the reviewer's concerns, and we sincerely thank the reviewer again for their thoughtful feedback.

Comment4: Original point 7: As we said previously, the Cas9-VP result doesn't show much, the Cas9 activator only induces expression by 8-fold, compared to a huge derepression on Nanog deletion. However, the result needs discussion as the removal of the element is hard to interpret. Is Nanog repressing GATA6 expression or just inhibiting the activity of the enhancer?

Response 4: We sincerely appreciate the meticulous review. Below is our detailed response to your concern regarding the regulatory relationship between Nanog, the DE region (187 kb upstream of GATA6), and GATA6 expression, identified using the dead Cas9 (dCas9)-Activator system and other experiments in this study.

1. The DE region is a functional enhancer of GATA6 (validated by multiple approaches): As shown in the manuscript, targeting the DE region with dCas9-activator upregulated GATA6 expression by over 8-fold, confirming its intrinsic enhancer activity. To verify its necessity, we knocked out the DE region via CRISPR-Cas9. New Figure 5D shows that DE deletion drastically attenuated the GATA6 upregulation induced by Nanog knockout—from over 100-fold (wild-type background) to ~10–30-fold—directly demonstrating that DE is indispensable for robust GATA6 induction upon Nanog depletion.

2. Reasons for the discrepancy in GATA6 upregulation levels: The lower GATA6 upregulation (8-fold) by dCas9-Activator compared to Nanog knockout (over 100-fold) stems from two key factors: 1. Limitations of transient transfection: The dCas9-Activator plasmid and guide RNA were delivered via transient

transfection, and incomplete transfection efficiency (not all cells expressed the fusion protein) led to insufficient DE activation; 2. Synergistic effects of other potential enhancers: Figure 5B indicates that GATA6 may harbor additional enhancers beyond DE. Nanog may repress GATA6 by synergistically inhibiting multiple enhancers (including DE and uncharacterized regions), so Nanog knockout relieves multi-layered repression, resulting in more dramatic GATA6 upregulation; 3. Other factors which may involve the Gata6 activation upon Nanog deletion are not involved by dCas9-activator.

3. Nanog represses GATA6 by inhibiting DE enhancer activity: Our data support the model that Nanog negatively regulates GATA6 expression by suppressing DE enhancer activity. The attenuated GATA6 upregulation upon DE deletion (following Nanog knockout) further confirms DE as a core target of Nanog-mediated repression. We have supplemented the Discussion section to clarify this regulatory logic. Thank you again for your constructive comments, which have significantly enhanced the rigor and clarity of our study.

Prof. Wensheng Zhang
Medical College of Soochow University
Cam-Su Genomic Resource Center
199 Renai Road
Jiangsu 215220
China

Dear Wensheng,

I am very pleased to accept your manuscript for publication in the next available issue of EMBO reports. Thank you for your contribution to our journal.

You may qualify for financial assistance for your publication charges - either via a Springer Nature fully open access agreement or an EMBO initiative. Check your eligibility: <https://link.springer.com/journal/44319/how-to-publish-with-us>

Kind regards,

Martina

>>> Please note that it is EMBO Reports policy for the transcript of the editorial process (containing referee reports and your response letter) to be published as an online supplement to each paper. If you do NOT want this, you will need to inform the Editorial Office via email immediately. More information is available here: <https://link.springer.com/partners/embo-press/editorial-policies#Peer%20review>